# An observational study of the effects of aerosols on diurnal variation of heavy rainfall and associated clouds over Beijing-Tianjin-Hebei

Siyuan Zhou[1,2,3], Jing Yang[1,2*], Wei-Chyung Wang[3], Chuanfeng Zhao[4], Daoyi Gong[1,2], Peijun Shi[1,2]

[1] State Key Laboratory of Earth Surface Process and Resource Ecology, Beijing Normal University, China

[2] Key Laboratory of Environmental Change and Natural Disaster, Faculty of Geographical Science, Beijing Normal University, China

[3] Atmospheric Sciences Research Center, State University of New York, Albany, New York 12203, USA

[4] College of Global Change and Earth System Science, Beijing Normal University, China

Submitted to ACP

Oct 2018

*Correspondence to: Jing Yang, State Key Laboratory of Earth Surface Process and Resource Ecology/ Key Laboratory of Environmental Change and Natural Disaster, Faculty of Geographical Science, Beijing Normal University, 19#Xinjiekouwai Street, Haidian District, Beijing 100875, China. E-mail: yangjing@bnu.edu.cn

**Abstract:** Our previous study found that the observed rainfall diurnal variation over Beijing-Tianjin-Hebei shows distinct signature of the effects of pollutants. Here we used the hourly rainfall data together with satellite-based daily information of aerosols and clouds to further investigate changes in heavy rainfall and clouds associated with aerosol changes. Because of the strong coupling effects, we also examined the sensitivity of these changes to moisture (specific humidity) variations. For heavy rainfall, three distinguished characteristics are identified: *earlier start time*, *earlier peak time*, and *longer duration*; and the signals are robust using aerosol indicators based on both aerosol optical depth and cloud droplet number concentration. In-depth analysis reveals that the first two characteristics occur in the presence of (absorbing) black carbon aerosols and that the third is related to more (scattering) sulfate aerosols and sensitive to moisture abundance. Cloud changes are also evident, showing increases in cloud fraction, cloud top pressure, the liquid/ice cloud optical thickness and cloud water path, and decrease in ice cloud effective radius; and these changes are insensitive to moisture. Finally, the mechanisms for heavy rainfall characteristics are discussed and hypothesized.

**Key words: aerosol, heavy rainfall, diurnal variation, cloud, Beijing-Tianjin-Hebei, observational study**

**1. Introduction**

Aerosols modify the hydrologic cycle through direct radiative and indirect cloud adjustment effects (IPCC, 2013). The direct effect, through absorbing and scattering solar radiation, leads to heating in the atmosphere (e.g. Jacobson 2001; Lau et al. 2006) and cooling on the surface (Lelieveld and Heintzenberg 1992; Guo et al. 2013; Yang et al., 2018), causing changes in atmospheric vertical static stability and subsequently modulation of rainfall (e.g. Rosenfeld et al. 2008). On the other hand, water-soluble aerosols serving as cloud condensation nuclei (CCN) affect the warm-rain and cold-rain processes through influencing the cloud droplet size distributions, cloud top heights and other cloud properties (Jiang et al., 2002; Givati and Rosenfeld 2004; Chen et al., 2011; Lim and Hong 2012; Tao et al., 2012). For Beijing-Tianjin-Hebei (BTH) the significant increase in pollution in recent decades has raised issues concerning aerosol-radiation-cloud-precipitation interactions. While the impact of aerosols on light rainfall or warm-rain processes is in general agreement among studies for this region (e.g., Qian et al., 2009), the uncertainties of the effects on heavy convective rainfall are still large (Guo et al., 2014; Wang et al., 2016).

The clouds that can generate heavy convective rainfall in BTH region usually contain warm clouds, cold clouds and mixed-phase clouds (e.g. Guo et al., 2015). Because the aerosol-cloud interactions in different types of clouds are distinct (Gryspeerdt et al., 2014b), aerosol indirect effect during heavy rainfall is more complicated than its direct effect (Sassen et al., 1995; Sherwood, 2002; Jiang et al., 2008, Tao et al., 2012). For warm clouds, by serving as CCN that nucleates more cloud droplets, aerosols can increase cloud albedo so called albedo effect or Twomey effect (Twomey, 1977), lengthen the cloud lifetime so called lifetime effect

(Albrecht, 1989), and enhance thin cloud thermal emissivity so called thermal emissivity effect (Garrett and Zhao, 2006). The above effects tend to increase the cloud microphysical stability and suppress warm-rain processes (Albrecht 1989; Rosenfeld et al. 2014). For cold clouds and mixed-phase clouds, many studies reported that the cloud liquid accumulated by aerosols is converted to ice hydrometeors above the freezing level, which invigorates deep convective clouds and intensifies heavy precipitation so called invigoration effect (Rosenfeld and Woodley, 2000; Rosenfeld et al., 2008; Lee et al. 2009; Guo et al. 2014). The Twomey effect infers that aerosols serving as CCN that increase the cloud droplets could reduce cloud droplet size within a constant liquid water path (Twomey, 1977). However, the opposite results of relationship between aerosols and cloud droplet effective radius were reported in observations (Yuan et al., 2008; Panicker et al., 2010; Jung et al., 2013; Harikishan et al., 2016; Qiu et al., 2017), which might be related with the moisture supply near the cloud base (Yuan et al., 2008; Qiu et al., 2017). Besides, the influence of aerosols on ice clouds also depends upon the amount of moisture supply (Jiang et al., 2008). Therefore, how the aerosols modify the heavy convective rainfall and associated cloud changes does not reach a consensus, particularly if considering the different moisture conditions.

Heavy convective rainfall over BTH region usually occurs within a few hours, thus studying on the relationship between aerosols and rainfall diurnal variation could deepen our understanding of aerosol effects on heavy rainfall. Several previous studies have found that aerosols are related to the changes of the rainfall diurnal variation in other regions (Kim et al., 2010; Gryspeerdt et al., 2014b; Fan et al., 2015; Guo et al., 2016; Lee et al., 2016). However, the above studies do not address the change of cloud properties and its sensitivity to different conditions of moisture supply. Although our recent work over BTH region (Zhou et al. 2018) attempted to remove the meteorological effect including circulation and moisture and found that the peak of heavy rainfall shifts earlier on the polluted condition, it only excluded the extreme moisture conditions and focused on aerosol radiative effect on the rainfall diurnal variation. Therefore, this study aims to deepen the previous study (Zhou et al., 2018) through investigating the following questions: (1) how do aerosols (including absorbing aerosols and scattering aerosols) modify the behaviors of the heavy rainfall diurnal variation (start time, peak time, duration and intensity)? And what is the role of moisture in them? (2) how do aerosols influence the associated cloud properties with inclusion of moisture? To solve above questions, we used aerosol optical depth (AOD) as a macro indicator of aerosol pollution and cloud droplet number concentration (CDNC) as a micro indicator of CCN served by aerosols respectively to compare the characteristics of heavy rainfall diurnal variation and cloud properties between clean and polluted conditions, and applied aerosol index (AI) to distinguish the associated different effects of absorbing aerosols and scattering aerosols. In addition, we used the specific humidity (SH) at 850 hPa as an indicator of moisture supply condition to investigate the possible effects of moisture on the rainfall and clouds and compared them with the effects of aerosols. The paper is organized as following: The data and methodology are introduced in Sect. 2. Section 3 addresses the relationship between aerosol pollution and diurnal variation of heavy rainfall, covering the distinct characteristics of heavy rainfall using AOD and CDNC; the different behaviors of heavy

rainfall diurnal variation along with different types of aerosols, and the comparison of heavy rainfall behaviors
influenced respectively by moisture and aerosols. Section 4 describes the concurrent changes of cloud
properties associated with aerosols and compares the possible influences of CDNC (CCN) and moisture on the
cloud properties. Section 5 gives the hypothesis about the mechanisms of aerosol effects on the heavy rainfall.
Conclusions and discussion will be given in Sect. 6.

## 2. Approach

### 2.1 Data

Four types of datasets from the year 2002 to 2012 (11 years) are used in this study, which include (1)
precipitation, (2) aerosols, (3) clouds, and (4) other meteorological fields.

#### 2.1.1 Precipitation

To study the diurnal variation of heavy rainfall, the gauge-based hourly precipitation datasets are used, which
were obtained from the National Meteorological Information Center (NMIC) of the China Meteorological
Administration (CMA) (Yu et al., 2007) at 2420 stations in China from 1951 to 2012. The quality control
made by CMA/NMIC includes the check for extreme values (the value exceeding the monthly maximum in
daily precipitation was rejected), the internal consistency check (wiping off the erroneous records caused by
incorrect units, reading, or coding) and spatial consistency check (comparing the time series of hourly
precipitation with nearby stations) [Shen et al., 2010]. Here we chose 176 stations in the plain area of BTH
region that are below the topography of 100 meter above sea level as shown in Fig.1, because we purposely
removed the probable orographic influence on the rainfall diurnal variation, which is consistent with our
previous work (Zhou et al., 2018). The record analyzed here is the period of 2002 to 2012. We selected heavy
rainfall days when the hourly precipitation amount is more than 8.0 mm/hour (defined by *Atmospheric*
*Sciences Thesaurus, 1994*). Here "a day" is counted from 8 LST to 8 LST next day (0 UTC to 24 UTC).

#### 2.1.2 Aerosols

In this study, we used two satellite data and one reanalysis data to investigate the aerosol optical amount and
distinguish the different aerosol types.
AOD is a proxy for the optical amount of aerosol particles in a column of the atmosphere and serves as the
macro indicator for the division of aerosol pollution condition in this study, which was obtained from MODIS
(Moderate Resolution Imaging Spectroradiometer) Collection 6 L3 aerosol product with the horizontal
resolution of 1°x1° onboard the Terra satellite (Tao et al., 2015). The quality assurance of marginal or higher
confidence is used in this study. The reported uncertainty in MODIS AOD data is on the order of (-0.02-10%),
(+0.04+10%) (Levy et al., 2013). The Terra satellite overpass time at the equator is around 10:30 local solar
time (LST) in the daytime, and the satellite data is almost missing when it is rainy during the overpass time.
As shown in Fig.3, the occurrence of selected heavy rainfall events in this study is mainly later than the
satellite overpass time. Therefore, the AOD used here represents the situation of the air quality in advance of
heavy rainfall appearance. Many studies have indicated the value of AOD is influenced by moisture condition,
which is aerosol humidification effect (Twohy et al., 2009; Altaratz et al., 2013). Hence, we comprehensively
analyzed the moisture effect on the rainfall and tried to remove the moisture effect from the relationship
between aerosols and rainfall/clouds.
The ultraviolet AI from Ozone Monitoring Instrument (OMI) on board the Aura satellite which was
launched in July 2004, is used for detecting the different types of aerosols in this study. The OMI ultraviolet
AI is a method of detecting absorbing aerosols from satellite measurements in the near-ultraviolet wavelength
region (Torres et al., 1998). The positive values of ultraviolet AI are attributed to the absorbing aerosols such
as smoke and dust while the negative values of AI stand for the non-absorbing aerosols (scattering aerosols)
such as sulfate and sea salt (Tariq and Ali, 2015). The near-zero values of AI occur when clouds and Rayleigh
scattering dominate (Hammer et al., 2018). Considering the near-zero values have more uncertainties, we only
compare the extreme circumstances of absorbing aerosols and scattering aerosols in this study. The horizontal
resolution of AI data is $1° \times 1°$ and it covers the period of 2005 to 2012.
MACC-II (Monitoring Atmospheric Composition and Climate Interim Implementation) reanalysis product
produced by ECMWF (the European Centre for Medium-Range Weather Forecasts), provided the AOD
datasets for different kinds of aerosols (BC, sulfate, organic matter, mineral dust and sea salt). MACC-II
reanalysis products are observationally-based within a model framework, which can offer a more complete
temporal and spatial coverage than observation and reduce the shortcomings of simulation that fail in
simulating the complexity of real aerosol distributions (Benedetti *et al*., 2009). The horizontal resolution of
MACC-II is also $1° \times 1°$ with the time interval of six-hour, and the results in the analysis of heavy rainfall show
consistent based on the daily mean values which is shown in the figures and morning values that before the
occurrence of heavy rainfall. MACC-II data covers the period of 2003 to 2012.
**2.1.3 Clouds**
Daily cloud variables, including cloud fraction (CF), cloud top pressure (CTP), cloud optical thickness (COT,
liquid and ice), cloud water path (CWP, liquid and ice) and cloud effective radius (CER, liquid and ice), were
obtained from MODIS Collection 6 L3 cloud product onboard the Terra satellite. The MODIS cloud product
combines infrared emission and solar reflectance techniques to determine both physical and radiative cloud
properties (Platnick et al., 2017). The validation of cloud top properties in this product has been conducted
through comparisons with CALIOP (Cloud-Aerosol Lidar with Orthogonal Polarization) data and other lidar
observations (Holz et al., 2008; Menzel et al., 2008), and the validation and quality control of cloud optical
products is performed primarily using in situ measurements obtained during field campaigns as well as the
MODIS Airborne Simulator instrument (https://modis-atmos.gsfc.nasa.gov/products/cloud). Consistent with
AOD, the measure of above cloud variables is before the occurrence of heavy rainfall.

In addition to the variables in MODIS cloud product, we also calculated CDNC using the liquid COT and CER in this product. CDNC is retrieved as the proxy for CCN and also the micro indicator for separating different aerosol conditions in this study. Currently, most derivations of CDNC assume that the clouds are adiabatic and horizontally homogeneous; CDNC is constant throughout the cloud's vertical extent, and cloud liquid water content varies linearly with altitude adiabatically (Min et al., 2012; Bennartz and Rausch, 2017). According to Boers et al. (2006) and Bennartz (2007), we calculated CDNC (unit: cm$^{-3}$) through:

$$\text{CDNC} = \frac{C_w^{1/2}}{k} \frac{10^{1/2}}{4\pi \rho_w^{1/2}} \frac{\tau^{1/2}}{R_e^{5/2}} \tag{1}$$

Where $C_w$ is the moist adiabatic condensate coefficient, and its value depends slightly on the temperature of the cloud layer, ranging from 1 to 2. 5 x $10^{-3}$ gm$^{-4}$ for a temperature between 0 ℃ and 40 ℃ (Brenguier, 1991). In this study, we calculated the $C_w$ through the function of the temperature (see Fig.1 in Zhu et al., 2018) at a given pressure that is 850 hPa. And we have tested the sensitivity of CDNC to the amount of $C_w$ and found it almost keeps the same when the $C_w$ changes from 1 to 2. 5 x $10^{-3}$ gm$^{-4}$. The coefficient k is the ratio between the volume mean radius and the effective radius, and varies between 0.5 and 1 (Brenguier et al., 2000). Here we used k = 1 for that we cannot get the accurate value of k and the value of k does not influence the rank of CDNC for the division of aerosol condition in this study. $\rho_w$ is cloud water density. $\tau$ and Re are the liquid COT and CER obtained from MODIS Collection 6 L3 cloud product with resolution of 1°x1°. To reduce the uncertainty of CDNC retrieval caused by the heterogeneity effect from thin clouds (Nakajima and King, 1990; Quaas et al., 2008; Grandey and Stier, 2010; Grosvenor et al., 2018), we selected the CF more than 80%, the liquid COT more than 4 and the liquid CER more than 4 μm when calculating the CDNC (Quaas et al., 2008).

### 2.1.4 Other meteorological data

In this study, wind, temperature, pressure and SH data, were obtained from the ERA-Interim reanalysis datasets with 1°x1°horizontal resolution and 37 vertical levels at six-hour intervals. The daily mean values of these variables are used in the study, and we also verified the results based on the morning values that before the occurrence of heavy rainfall. ERA-Interim is a global atmospheric reanalysis produced by ECMWF, which covers the period from 1979 to near-real time (Dee et al., 2011). The SH, which stands for the water vapor content, serves as the indicator of moisture supply condition in this study.

### 2.2 Methodology

We used both station data of gauge-based precipitation and gridded data including aerosols, clouds and other meteorological variables. Gridded datasets in this study were downloaded with the horizontal resolution of 1°×1°, which are consistent with the resolution of MODIS L3 products. To unify the datasets, we interpolated all the gridded datasets onto the selected 176 rainfall stations using the average value in a 1°×1° grid as the

background condition of each rainfall station, i.e., the stations in the same 1°×1° grid have the same aerosol, cloud and meteorological conditions.

### 2.2.1 Selection of sub-season and circulation

Consistent with our previous work, we focused on the early summer period (1 June to 20 July) which is before the large-scale rainy season start, in order to remove the large-scale circulation influence and identify the effect of aerosols on local convective precipitation because BTH rainfall during this period is mostly convective rainfall (Yu et al., 2007) with heavy pollution (Zhou et al., 2018). And to unify the background atmospheric circulation, we only selected the rainfall days with southwesterly flow, which is the dominant circulation accounting for 40% of total circulation patterns over the BTH region during early summer (Zhou et al., 2018).

### 2.2.2 Classification of clean/polluted cases and moisture conditions

With the circulation of southwesterly, we used two indicators to distinguish the clean and polluted conditions from macro and micro perspectives, which are AOD and CDNC. The $25^{th}$ and $75^{th}$ percentiles of AOD/CDNC of the whole rainfall days are used as the thresholds of clean and polluted conditions, and the values are shown in Tab.1. There are 514 cases of heavy rainfall on the polluted days and 406 cases of that on the clean days when using AOD, and 630/716 cases on the polluted/clean condition when using CDNC (Fig. 3).

The absorbing aerosols are detected using the positive values of AI that is named as absorbing aerosol index (AAI) here, and we can retrieve the scattering aerosol index (SAI) using the negative values of AI. AAI and SAI are also divided into two groups using the threshold of $25^{th}/75^{th}$ percentile as shown in Tab.1. We used AAI/SAI more than $75^{th}$ percentile as the extreme circumstances of absorbing/scattering aerosols to compare their impacts on the heavy rainfall. The sample numbers are 375 and 550 respectively for the extreme AAI and SAI cases. Using the same method, we chose cases with more BC/sulfate when the AOD of BC/sulfate is larger than the $75^{th}$ percentile of itself in all rainy days, and cases with less BC/sulfate when that is less than the $25^{th}$ percentile of itself in the same situation. Accordingly, we selected 459 heavy rainfall cases with more BC and 274 cases with less BC. Similarly, 361 cases with more sulfate and 419 cases with less sulfate were selected (Fig. 6).

The SH at 850 hPa is used as the indicator of moisture supply under the cloud base. We chose wet cases when the SH on that day is larger than $75^{th}$ percentile of the whole rainy days, and chose dry cases when SH on that day is less than the $25^{th}$ percentile of the whole rainy days (the thresholds are shown in Tab. 1).

### 2.2.3 Statistical analysis

We adopted the probability distribution function (PDF) to compare the features of heavy rainfall and cloud variables on different conditions of aerosols, through which we can understand the changes of rainfall/cloud properties more comprehensively. The numbers of bins we selected in the study have been all tested for better representing the PDF distribution. Student's t-test is used to examine the significance level of differences

between the different groups of aerosol conditions. The differences between any two groups that have passed
95% statistical confidence level are considered significant. And two variables are considered correlated when
the correlation coefficient is more than 0.5 or less than -0.5.

**3. Changes of heavy rainfall**
In this study, we used two indicators (AOD and CDNC) to identify the aerosol pollution. AOD is usually
used as the macro indicator of aerosol pollution, which represents the optical amount of aerosol particles.
However, AOD is not a proper proxy for CCN (Shinozuka et al., 2015), but the property of aerosols serving as
CCN should be considered because aerosol-cloud interaction plays an indispensable role on changing rainfall
diurnal variation. Therefore, here we applied the retrieved CDNC as the indicator of CCN (Zeng et al., 2014;
Zhu et al., 2018).
We first investigated the value distribution of AOD and CDNC over the BTH region. Figure 2a&b shows
the PDFs of AOD and CDNC on the non-rainfall days, rainfall days and heavy rainfall days respectively. The
spectral distributions of AOD on different conditions are quite similar that the ranges are all between 0-5 and
the peaks occur at around 1.2 (Fig. 2a). In contrast, CDNC shows different ranges between different
conditions, that it ranges from around 20 $cm^{-3}$ to 500 $cm^{-3}$ on the rainfall days and non-rainfall days while
from around 30 $cm^{-3}$ to 420 $cm^{-3}$ on the heavy rainfall days. Besides, the proportion of low CDNC is quite
high on the non-rainfall days (Fig. 2b). The averaged CDNCs on the non-rainfall days, rainfall days and heavy
rainfall days are 54.70, 72.92, and 68.66 $cm^{-3}$ respectively. According to the above results, the range of AOD
remains similar on the heavy rainfall days while the range of CDNC is shortened, probably because the cloud
droplets become larger before heavy rainfall so that the number concentration becomes less. Therefore, to
obtain comparable samples, we use percentile method to select respective clean and polluted cases based on
above two indicators and compare the characteristics of heavy rainfall. Hence the heavier pollution means
larger optical amount of aerosols measured by AOD, and more aerosols that could serve as CCN measured by
CDNC.
**3.1 Characteristics**
Our previous study (Zhou et al. 2018) has reported the distinct peak shifts of rainfall diurnal variation between
clean and polluted days using the indicator of AOD over the BTH region during early summer. Similar with
our previous study, the PDF of the heavy rainfall peak time shows that the maximum of rainfall peak is about
two hours earlier on the polluted days (20:00 LST) than that on the clean days (22:00 LST) (Fig. 3a). To
comprehensively recognize the changes of rainfall diurnal variation associated with air qualities, here we
examined the PDF of the start time, the duration and the intensity besides the peak time of heavy rainfall.
As shown in Fig. 3a, the start time of heavy rainfall exhibits a significant advance on the polluted days. The
secondary peak on the early morning is ignored here because the early-morning rainfall is usually associated

with the mountain winds (Wolyn et al., 1994; Li et al., 2016) and the nighttime low-level jet (Higgins et al., 1997; Liu et al., 2012) that is beyond the scope of this study. The time for the maximum frequency of heavy rainfall initiation is around 6 hours earlier on the polluted days, shifting from around 0:00 LST on the clean days to the 18:00 LST (Fig. 3a). Regarding the rainfall durations, the average persistence of heavy rainfall on polluted days is 0.8 hours longer than that on clean days (Tab. 2). According to the PDF shown as in Fig. 3a, the occurrence of short-term precipitation ($\leq$6 hours, Yuan et al., 2010) decreases while that of long-term precipitation (>6 hours, Yuan et al., 2010) increases. The intensity of hourly rainfall exhibits a non-significant increase on the polluted days.

The distinct behaviors of heavy rainfall diurnal variation between clean and polluted days have been well demonstrated using the indicator of AOD. Using CDNC as the indicator of CCN, the above-mentioned results are also significant, as shown in Fig. 3b. The start time and peak time of heavy rainfall on the polluted condition also show significant advances compared with that on the clean condition, with the average advances of 1.4 hours and 3.0 hours respectively (Tab. 2). The duration of heavy rainfall on the polluted condition is also prolonged, which is 2.2 hours longer in average (Tab. 2). Similar with the results based on AOD, the difference of rainfall intensity between clean and polluted conditions using CDNC does not pass the 95% statistical confidence level as well.

Hence, the results using either AOD or CDNC show that the start and peak time of heavy rainfall occur earlier and the duration becomes longer under pollution, although there are some quantitative differences between the two indicators. We found the AOD and CDNC only have a non-significant positive correlation, which denotes that the selected cases could be different between using AOD and CDNC. The cases of heavy rainfall using CDNC seem more extreme, because the rainfall behaviors exhibit more evident changes using CDNC than using AOD. The result differences between the two indicators might be attributed to the non-linear relationship between CCN and aerosol pollution (e.g., Jiang et al., 2016), the misdetection of AOD when the humidity is high (Boucher and Quaas, 2012), the calculation uncertainty of CDNC, and the sampling differences between AOD and CDNC. Since the two indicators represent aerosols from the different perspectives, we cannot identify which one is more reliable. Because the change of rainfall intensity is not significant using either AOD or CDNC, the following analysis only focuses on studying the start time, peak time and duration of heavy rainfall along with aerosol pollution.

## 3.2 Sensitivities to aerosol types

Using the indicator of AI, we further investigated the distinct behaviors of heavy rainfall diurnal variation related to absorbing aerosols and scattering aerosols respectively. The PDF of start time, peak time and duration of heavy rainfall under the extreme circumstances of absorbing aerosols and scattering aerosols are compared in Fig. 4. Here, we briefly named the days with extreme large amount of absorbing aerosols as absorbing aerosol days and with more scattering aerosols as scattering aerosol days. The start time of heavy rainfall on absorbing aerosol days shows a significant earlier compared with that on scattering aerosol days

(Fig. 4a), with 0.7 hours advance in average (Tab. 3). Similarly, the rainfall peak time also shows earlier on
absorbing aerosol days (Fig. 4b), with an average advance of 1.6 hours (Tab. 3). The rainfall duration on
scattering aerosol days shows longer than that on absorbing aerosol days, which are 6.0 hours and 5.0 hours
respectively in average (Tab. 3). All the above-mentioned differences between the two groups have passed 95%
statistical confidence level. The results indicate that the absorbing aerosols and scattering aerosols may have
different or inverse effects on the heavy rainfall that absorbing aerosols may generate the heavy rainfall in
advance while the scattering aerosols may delay and prolong the heavy rainfall.
To further verify the different behaviors of heavy rainfall diurnal variation associated with two different
types of aerosols, we purposely re-examine the above-mentioned phenomena using BC/sulfate that can
represent typical absorbing/scattering aerosols over the BTH region. BC has its maximum center over BTH
region (Fig. 5a) and our previous study has indicated that the radiative effect of BC low-level warming may
facilitate the convective rainfall generation (Zhou et al., 2018). The percentage of sulfate is also large over the
BTH region (Fig. 5b) and the sulfate is one of the most effective CCN that influences the precipitation in this
region (Gunthe et al., 2011). Accordingly, we selected the cases with different amounts of BC and sulfate
AOD to compare their roles on the diurnal variation of heavy rainfall. The methods have been described in
Sect. 2.2.2. The PDF of the start time, peak time and duration of heavy rainfall in the cases with more/less
amount of BC are shown in Fig. 6a, respectively. The most striking result is that the maximum frequency of
rainfall start time in the more BC cases evidently shifts earlier (Fig. 6a). Meanwhile, the mean peak time in
the more BC cases shows 1.1 hour earlier than that in the less BC cases (Tab. 3). And the duration of heavy
rainfall is slightly shortened by the averaged 0.2 hours in the more BC cases. The features in more BC cases
are consistent with the above results of absorbing aerosols. In contrast, when the sulfate has higher amount,
the mean start time of rainfall is delayed by 0.5 hours, while the duration shows a significant increase by 1.5
hours in average (Tab. 3). The behaviors in the more sulfate cases also exhibit similar with the above results
of scattering aerosols, except for the peak time that shows later in the scattering aerosol cases but a little
earlier in the more sulfate cases (Tab. 3).
**3.3 Influence of moisture**
Moisture supply is an indispensable factor for the precipitation formation, and it also has an important impact
on AOD (Boucher and Quaas, 2012). Since the southwesterly circulation can not only transport pollutants but
also plenty of moisture to the BTH region (Wu et al., 2017), more pollution usually corresponds to more
moisture for the BTH region (Sun et al., 2015) so that it is hard to completely remove the moisture effect on
the above results in the pure observational study. Here we attempt to recognize the moisture effect on the
heavy rainfall to further understand the above aerosol-associated changes. Because the moisture supply for
BTH is mainly transported via low-level southwesterly circulation, we purposely used the SH at 850 hPa as
the indicator of moisture condition.
Using the similar percentile method with polluted/clean days, we compared the heavy rainfall
characteristics in the more humid (more than 75[th] percentile) and the less humid (less than 25[th] percentile)
environments regardless of the aerosol condition, as shown in Fig. 7a. The results show that the start time of
heavy rainfall is delayed by 0.9 hours, the peak time is 0.6 hours earlier and the duration is prolonged by 2.0
hours in average in the more humid environment, which is similar with the results of the more sulfate cases.
Besides, the same results are obtained using different moisture indicator, e.g. the 850 hPa absolute humidity.
These results indicate the advance of heavy rainfall start time on the polluted days is not caused by more
moisture supply, while the longer duration and earlier peak in the more sulfate cases might be related to the
increased moisture supply. To further identify the role of sulfate, we tested the sensitivities of the results
associated with sulfate when limiting the moisture condition. In the dry and intermediate cases (SH less than
25[th] percentile and SH between 25[th] -75[th] percentiles), the heavy rainfall still shows later start time, earlier
peak and significant longer duration with the increase of sulfate, while the change of peak time is not
significant in the dry cases and that of start time is not significant in the intermediate cases; in the high
moisture cases (SH more than 75[th] percentile), it shows earlier peak and shorter duration in the more sulfate
cases while the change of start time is not significant. Therefore, we suppose that the impact of sulfate
aerosols on the heavy rainfall is sensitive to moisture, and notably the sulfate could contribute to the longer
duration in the polluted cases when it is relatively dry.
We also investigate the distributions of moisture and rainfall behaviors in the clean and polluted cases
respectively using AOD and CDNC (Fig. 7 b&c). The results show that the relationship between moisture and
rainfall start time/peak time/duration is not linear. Using either AOD or CDNC, the distribution of SH exhibits
a slight increase in the polluted cases, indicating that the polluted cases have the more moisture than the clean
cases which is particularly well shown using AOD. However, when fixing the moisture at a certain range
especially at the relative dry condition, we can detect the similar phenomena of earlier start/peak time and
longer duration in the polluted cases. For example, when the amount of 850 hPa SH is between 8-12 g/kg, the
start &peak time in the polluted cases show significant earlier and the duration exhibits slightly increased
compared with that in the clean cases using either AOD or CDNC. To further clarify the characteristics of
heavy rainfall associated with pollution, we removed the samples with high SH (SH more than 75[th] percentile)
and found that the results in section 3.1 remain the same, that when SH is less than 12.95 g/kg (75[th] percentile),
the start/peak time of heavy rainfall is also in advance and the duration is still prolonged with the increase of
AOD/CDNC (Fig. 8).
The above results indicate that the advance of heavy rainfall start in the polluted cases is independent of
moisture, while the advance of peak time and longer duration might be related to the moisture effect. For the
peak time of heavy rainfall, although the results of BC, sulfate and moisture show consistent advance, we
suppose the role of BC (absorbing aerosols) might be dominant compared with that of sulfate or moisture
because the change of peak time in the former analysis is much larger (Tab. 3). Both sulfate and moisture may
contribute to the longer duration of heavy rainfall (Fig. 6b&7a), but the role of sulfate seems sensitive to the
moisture condition, that the duration in the more sulfate cases is longer when the moisture condition is
relatively dry while becomes shorter when it is extremely wet. Because we cannot completely separate the
sulfate and moisture, we are not quite clear about their individual roles at present. Overall, under the condition
of removing the extremely high moisture cases, the earlier start/peak time and longer duration of heavy
rainfall associated with aerosol pollution are significant and irrespective of moisture.

**4. Changes of clouds**
To understand the cloud effect of aerosols during heavy rainfall diurnal variation, we need to recognize the
associated cloud characteristics on the clean and polluted conditions. The cloud properties we used were
obtained from satellite product that were measured at the same time with aerosols before the occurrence of
heavy rainfall. The differences of cloud features were examined in both macroscopic (including CF, CTP,
COT and CWP) and microscopic properties (including CER) between the clean and polluted conditions based
on AOD and CDNC respectively.
**4.1 Characteristics**
Using AOD as the macro aerosol indicator, as shown in Fig. 9, the PDF distribution of CF shows that the CF
on the polluted condition is evidently larger than that on the clean condition. The average CF is 62.8% on the
clean condition, and 89.3% on the polluted condition (Tab. 4), which is increased by 26.1%. The average CTP
on the polluted condition is 487.3 hPa, which is larger than 442.3 hPa on the clean condition, indicating that
the cloud top height is lower on the polluted days. The COT, CWP and CER were further analyzed for the
liquid and ice portions of clouds as shown in Fig. 9. Both liquid and ice COT on the polluted condition exhibit
significant increases compared with that on the clean condition. The mean amount of liquid COT is increased
by 3.1 and ice COT increases by 6.2 (Tab. 4). Similar with COT, the amounts of liquid and ice CWP also
increase under pollution, which increase by 33.6 $g/m^2$ and 88.2 $g/m^2$ respectively. In addition, the liquid CER
is increased by 0.8 μm and the ice CER is decreased by 2.8 μm on the polluted days. The differences of above
cloud properties between clean and polluted cases have all passed the 95% statistical confidence level.
Using CDNC as the micro aerosol indicator, the above-mentioned changes of cloud properties are
consistent with that using AOD, except for liquid CER (Fig. 9). Since the calculation method of CDNC is not
independent on the liquid COT and liquid CER, we would not directly compare the results of liquid COT and
CER based on CDNC with those based on AOD here. But according to other variables that are independent of
the CDNC calculation, we found the cases with more CDNC are accompanied with the increase of CTP, ice
COT and liquid & ice CWP, which increase by 32.8 hPa, 24.4, 215.8 $g/m^2$ and 370.9 $g/m^2$ respectively (Tab 4)
and all of which are consistent with the results based on AOD. The CER of ice clouds also shows a consistent
decrease by 8.8 μm on the polluted condition based on CDNC. We noticed that the changes of
COT/CWP/CER for both liquid and ice based on CDNC are much larger than that based on AOD, which
indicates that these cloud properties might be more sensitive to the indicator of CDNC rather than AOD.
According to the above comparison, the concurrent changes of cloud properties along with heavy rainfall
diurnal variation show consistent results using the two aerosol indicators (AOD and CDNC). The pollution
corresponds to the increase of CF, ice COT, liquid and ice CWP, but the decrease of cloud top height (the
increase of CTP corresponds to the decrease of cloud top height) and ice CER. The liquid COT and liquid
CER are also increased with the enhanced pollution in the AOD analysis. Besides, these above results exhibit
significant when we limited the moisture to the dryer condition (SH less than 25[th] percentile) or intermediate
condition (SH more than 25[th] percentile and less than 75[th] percentile). When the moisture is higher (SH more
than 75[th] percentile), the change of CTP does not show significant based on CDNC.
For these results, we made the following speculation: First, the CF, liquid & ice COT and CWP increase
with pollution, might because the aerosols serving as CCN can nucleate a larger number of cloud droplets and
accumulate more liquid water in the cloud thus increase the CF, COT and CWP. Second, the CTP increases
under pollution using both AOD and CDNC, which denotes the decrease of the cloud top height, might
because the earlier start of the precipitation process (Fig. 3) inhibits the vertical growth of clouds. Third, the
ice CER decreases under pollution using either AOD or CDNC, probably because the increased cloud droplet
number leads to more cloud droplets transforming into ice crystals and causes the decrease of ice CER
(Chylek et al., 2006; Zhao et al., 2018; Gryspeerdt et al., 2018). However, the results of liquid CER might
have uncertainties. The liquid CER is increased when AOD increases (Fig. 9), which might be related to the
aerosol humidification effect, the misdetection of AOD and cloud water, and also might result from the earlier
formation of the clouds and heavy rainfall on the polluted days. Since we cannot distinguish the liquid part of
mix-phased clouds from liquid (warm) clouds in the observation, the above changes of liquid cloud properties
might come from both the liquid (warm) clouds and the liquid part of mixed-phase clouds. Likewise, the
above-mentioned changes of ice cloud properties might come from both ice (cold) clouds and the ice part of
mixed-phase clouds. Currently the detailed physical processes of cold clouds and mixed-phase clouds have
been not clarified yet, including the diffusional grow, accretion, riming and melting process of ice
precipitation (Cheng et al., 2010), which needs numerical model simulations to be further explored.
**4.2 Sensitivities to CCN and moisture**
Section 3.3 has shown that the diurnal variation of heavy rainfall with more moisture supply is similar with
the changes of heavy rainfall with more sulfate aerosols. We assume that the moisture under the cloud base
and the sulfate serving as CCN both influence the cloud properties (Yuan et al., 2008; Jiang et al., 2008; Jung
et al., 2013; Qiu et al., 2017). To identify the effect of CCN on clouds and its sensitivity to moisture, using
CDNC to represent CCN, we purposely investigated the changes of above cloud properties on the different
conditions of the CDNC and the low-level moisture (850hPa SH) respectively.
We categorized all cases of heavy rainfall into four groups, which are (1) clean and dry, (2) polluted and
dry, (3) clean and wet, (4) polluted and wet, and checked the changes of above cloud properties, as shown in
Tab. 5. To retrieve the comparable samples, here "clean/polluted" refers to the CDNC on that day less/more

than $25^{th}/75^{th}$ percentile of the CDNC among the heavy rainfall days, and similarly, the "dry/wet" refers to the SH on that day less/more than $25^{th}/75^{th}$ percentile of itself among the heavy rainfall days. The average CDNC is 68.58 cm$^{-3}$ on the dry condition and 68.56 cm$^{-3}$ on the wet condition, and the average SH is 11.3 g/kg and 11.8 g/kg on the clean and polluted conditions respectively, thus we can consider the CDNC or SH remains the same when the other condition changes. We made the significant test of differences between group 1 and 2, group 1 and 3, group 2 and 4, group 3 and 4. Because the CF is fixed above 80% when calculating the CDNC (see in Sect. 2.1.3), here the selected groups all belong to the condition of higher CF.

Comparing the results of group 1 and 2, which are both on the dry condition, we can identify the influence of CDNC on the cloud properties, which stands for the effect of CCN. The changes of these cloud variables are the same as that in Sect. 4.1, that the CF, ice COT and liquid & ice CWP are increased on the polluted condition, while the cloud top height and ice CER are decreased based on CDNC. Among these variables, the ice COT and liquid & ice CWP are especially larger on the polluted condition, which are 5-6 times larger than that on the clean condition (Tab. 5). On the wet condition, comparing the group 3 and 4, the changes are similar that the CF, ice COT and liquid & ice CWP are increased and the ice CER are decreased but the change of CTP becomes not significant. However, the changes of these variables on the dry condition are evidently enhanced than that on the wet condition, which indicates these cloud properties might be more sensitive to CDNC on the dry condition. The above comparisons indicate that with the increase of CDNC (CCN), the CF, ice COT and liquid & ice CWP are increased while the ice CER is decreased regardless of the moisture amount. Although the comparisons of liquid COT and liquid CER based on CDNC are meaningless since the CDNC is calculated by the two variables, we infer that the increase of liquid COT and the decrease of liquid CER (Tab. 5) might be not completely caused by CDNC calculation but the natural effect of CCN.

Comparing the results of group 1 and 3, we can get the changes of cloud properties related only to moisture on the same clean condition. A common feature is that CTP, COT and CWP both for liquid and ice exhibit increases along with the increase of moisture. Compared with the CTP on the clean and dry condition, it increases on both polluted & dry condition (group 2) and clean & wet condition (group 3), but on the former condition its increase is larger, which indicates the influence of moisture on CTP might be secondary compared to the CDNC (CCN) effect. Similarly, comparing the COT/CWP in group 2 and 3, the increases of COT and CWP both for liquid and ice in group 2 are 3-6 times larger than that in group 3, which indicates that the influences of moisture on COT and CWP may not overcome the influence of CCN. With the increase of moisture, the change of liquid CER is not significant on the same clean condition, but the ice CER is significantly decreased. On the polluted condition, comparing group 2 and 4, we found the COT and CWP both for liquid and ice on the wet condition are evidently smaller than that on the dry condition, which indicates that increasing the moisture might partly compensate for the influence of CDNC (CCN) on COT/CWP.

The results above indicate that both CDNC (CCN) and moisture have impacts on cloud properties. They

both contribute to the increase of CF, COT and CWP, in which the influence of CDNC (CCN) on COT and CWP are significantly larger than moisture. The increase of either CDNC or moisture corresponds to the increase of CTP. But when the CDNC and moisture increase simultaneously, the CTP becomes smaller. Both CDNC and moisture correspond to the significant decrease of ice CER, while only CDNC corresponds to the decrease of liquid CER and that might be ascribed to the calculation method of CDNC. To reduce uncertainties, we have tested the SH at different levels (e.g., 700 hPa and 800 hPa) and different moisture indicator (e.g. absolute humidity) to verify these results, and found most cloud variables show the similar changes with above except for the CTP and the liquid CER, which indicates the changes of CTP and liquid CER are more sensitive and have larger uncertainties. Since the behaviors of cloud changes are similar along with the increase of either CDNC (CCN) or moisture but more sensitive to the former, the results in Sect. 4.1 might actually reflect the combined effect of CCN and moisture, and the aerosol effect on these cloud properties might be dominant on the polluted days.

Therefore, combining with the results in Sect. 3.3, although we cannot completely separate the aerosols and moisture, the CCN is assumed to play a vital role on the clouds and precipitation especially in a relatively dry environment. In the relatively wet environment, the CCN might have some inhibitory effect since the duration of heavy rainfall is shorter with the increase of sulfate when it is extremely wet, and the changes of cloud features along with the CDNC increase are smaller on the wet condition. Due to the limitations of observational study, we currently cannot figure out the respective roles of aerosols and moisture.

**5. Hypothesis**

According to all the above results, we have made hypotheses about the aerosol effects on the heavy rainfall over the BTH region. In Sect. 3.1 we found that the heavy rainfall has earlier start and peak time, and longer duration on the polluted condition. And afterwards, the earlier start of rainfall under pollution was found related to absorbing aerosols mainly referring to BC (Fig. 4a&6a). We also compared the effect of BC on the associated clouds. Figure 10a shows the CF larger than 90% rarely occurs in the more BC environment, which might be associated with the semi-direct effect of BC (Ackerman, 2000) or estimated inversion strength and BC co-vary. This result indicates the influence of BC on the heavy rainfall in Fig. 6a is mainly due to the radiative effect rather than the cloud effect. The mechanism of BC effect on the heavy rainfall can be interpreted by our previous study (Zhou et al., 2018) as: BC absorbs shortwave radiation during the daytime and warms the lower troposphere at around 850 hPa, and then increases the instability of the lower to middle atmosphere (850-500hPa) so that enhances the local upward motion and moisture convergence. As a result, the BC-induced thermodynamic instability of the atmosphere triggers the occurrence of heavy rainfall in advance. Thus, the low-level heating effect of BC might play a dominant role in the beginning of rainfall especially before the formation of clouds during the daytime.

The delayed start of heavy rainfall with scattering aerosols in Fig. 4a and more sulfate in Fig. 6b is
consistent with many studies that both the radiative effect and cloud effect of sulfate-like aerosols could delay
or suppress the occurrence of rainfall (Guo et al., 2013; Wang et al., 2016; Rosenfeld et al. 2014). Sulfate-like
aerosols as scattering aerosols could prevent the shortwave radiation from arriving at the surface thus cool the
surface and stabilize the atmosphere, which suppresses the rainfall formation (Guo et al., 2013; Wang et al.,
2016). Sulfate-like aerosols serving as CCN can also suppress the rainfall by cloud effect through reducing the
cloud droplet size and thus suppressing the collision-coalescence process of cloud droplets (Albrecht 1989;
Rosenfeld et al. 2014). Figure 10b does shows that in contrast with BC, the CF larger than 90% is
significantly increased in the more sulfate environment, which indicates the sulfate-like aerosols might have
more evident influence on the clouds and subsequently the rainfall changes associated with sulfate are
probably due to the cloud effects. Another significant feature is the longer duration of heavy rainfall in the
scattering aerosol cases, more sulfate cases and high moisture cases (Fig 4c, 6b&7a). We speculate that the
postponed start of heavy rainfall is mainly due to the effect of sulfate-like aerosols, while the longer duration
is caused by both the cloud effect of sulfate-like aerosols and the increased moisture supply, because
increasing either CCN or the moisture supply can increase cloud water (Sect. 4.2), which could lead to the
longer rainfall duration. To further investigate the mechanism of longer duration, we need the assistance of
numerical model simulations in the future work.
Accordingly, we speculate that the earlier start time of heavy rainfall related to absorbing aerosols (BC) is
due to the radiative heating of absorbing aerosols, while the longer rainfall duration is probably caused by
both the cloud effect of sulfate-like aerosols and the increased moisture supply. As a summary we use a
schematic diagram (Fig. 11) to illustrate how aerosols modify the heavy rainfall over the BTH region. On one
hand, BC heats the lower troposphere, changing the thermodynamic condition of atmosphere, which increases
the upward motion and accelerates the formation of clouds and rainfall. On the other hand, the increased
upward motion transports more sulfate-like particles and moisture into the clouds so that more CCN and
sufficient moisture increase the cloud water, thus might prolong the duration of rainfall. As a result, the heavy
rainfall over BTH region in southwesterly shows earlier start and peak time, and longer duration might due to
the combined effect of aerosol radiative effect, aerosol cloud effect as well as the moisture effect. To further
distinguish the individual effect, we need to conduct numerical model simulations in our future study.

**6. Conclusions and discussion**
**6.1 Conclusions**
Using the gauge-based hourly rainfall records, aerosol and cloud satellite products and high temporal
resolution reanalysis datasets during 2002-2012, this study investigated the different characteristics of heavy
rainfall in the diurnal time scale on the clean and polluted conditions respectively. Based on macro and micro

aerosol indicators that are AOD from MODIS aerosol product and calculated CDNC from MODIS cloud product, we found three features of heavy rainfall changing associated with aerosols that the rainfall start and peak time occur earlier and the duration becomes longer. The quantitative differences exist between the two indicators, i.e., the statistic differences of above features between clean and polluted conditions are 0.7, 1.0, 0.8 hours based on AOD and 1.4, 3.0, 2.2 hours based on CDNC.

The different relationships of absorbing and scattering aerosols to the diurnal shift were also distinguishable using ultraviolet AI from OMI and reanalysis AOD of two aerosol types (BC and sulfate). The absorbing aerosols (BC) correspond to the earlier start and peak time of heavy rainfall, while the scattering aerosols (sulfate) correspond to the delayed start time and the longer duration. Considering that the moisture has indispensable influence on the rainfall, the role of moisture (SH at 850 hPa) on the heavy rainfall is also investigated, which shows similar with the scattering aerosols (sulfate). Further analysis indicates the duration of heavy rainfall is prolonged in the presence of more sulfate on the relatively dry condition but is shortened on the extremely wet condition.

By comparing the characteristics of cloud macrophysics and microphysics variables, using both AOD and CDNC we found the CF, ice COT, liquid and ice CWP are increased on the polluted condition, but the cloud top height and the ice CER are reduced. Liquid COT and liquid CER are also increased in AOD analysis. Comparing the influences of CDNC which represents CCN and SH at 850 hPa which represents moisture condition respectively on these cloud variables, the cloud properties show similar changes with the increase of CDNC and moisture, but seem more sensitive to the CDNC (CCN), e.g., the liquid & ice COT and CWP increase more in the environment of high CDNC than that of high SH.

According to these results, we speculate that both aerosol radiative effect and cloud effect have impacts on the diurnal variation of heavy rainfall in the BTH region. The heating effect of absorbing aerosols especially BC increases the instability of the lower to middle atmosphere so that generates the heavy rainfall occurrence in advance. And the increased moisture supply and increased aerosols which nucleate more cloud droplets and accumulate more liquid water in clouds, leading to the longer duration of heavy rainfall.

**6.2 Discussion**

In this study we used two aerosol indicators, AOD and CDNC, which discriminates the pollution levels for different purposes. AOD is a good proxy for the large-scale pollution level, but it stands for the optical feature of aerosols and cannot well represent CCN when we focused on the aerosol-cloud interaction (Shinozuka et al., 2015). CDNC is a better proxy for CCN compared with AOD, which facilitates the study on the cloud changes associated with aerosol pollution. But the retrieved CDNC has larger uncertainties. First, the assumptions in the calculation of CDNC are idealized that CDNC is constant with height in a cloud and cloud liquid water increases monotonically at an adiabatic environment (Grosvenor et al., 2018), but the target of this study is the convective clouds with rainfall that may be not consistent with the adiabatic assumption. Second, as indicated

by Grosvenor et al. (2018), the uncertainties in the pixel-level retrievals of CDNC from MODIS with 1°x1°
spatial resolution can be above 54%, which come from the uncertainties of parameters and the original COT
and CER data using in the calculation, and also the influence of heterogeneity effect from thin clouds. To
reduce the influence of heterogeneity effect as much as possible, we have attempted to limit the conditions of
CF, liquid COT and CER when calculating CDNC in the study. Besides, this study primarily focuses on the
relative changes of CDNC, which may be also influenced by the potential systematic biases in the CDNC
calculation, but actually reduced the uncertainties of absolute values. Another problem of CDNC in this study
is that the CDNC is actually influenced by updraft velocity when we use CDNC to represent CCN since both
CCN and updraft velocity could contribute to aerosol activation and increase CDNC (Reutter et al., 2009). We
used the vertical velocity at 850 hPa obtained from ERA-interim reanalysis data to investigate the relationship
between CDNC and updraft, and the results show that there is no significant correlation found between CDNC
and vertical velocity (among all cases or just polluted cases), although the vertical velocity is larger in the
polluted cases. We also checked the results of rainfall based on CDNC when limiting the vertical velocity to a
certain range (less than $25^{th}$ percentile, $25^{th} - 75^{th}$ percentile or more than $75^{th}$ percentile), and the main
conclusion did not change. Besides, the increase of vertical velocity on the polluted days during daytime we
think is related to the pollutants (Zhou et al., 2018), which means the pollutants including the aerosols serving
as CCN and the updraft are co-varied and they might be the cause and effect for each other. Therefore, we
suppose the CDNC could stand for CCN to a certain extent although it might be partially dependent of updraft
velocity.

In addition to AOD and CDNC, we also applied ultraviolet AI and AOD of BC/sulfate to identify different

types of aerosols. We found that the AI has a weak positive correlation with AOD from MODIS, which
indicates the results on absorbing aerosol days might represent the results on polluted days if identified by
AOD. To avoid the uncertainty, we re-examined the results using AI when removing the polluted cases
identified by AOD, and found the major results are not changed. The comparisons of BC/sulfate AOD cases
also have uncertainties because they are retrieved from MACC reanalysis data. Although the above four
indicators have their own uncertainties, currently we cannot find more reliable datasets in a long-term
observational record. The major findings using these four indices could well identify the changes of rainfall
and clouds accompanied with aerosols, but are insufficient to clarify the aerosol effect on clouds and
precipitation.

This study has clearly identified the relationship of the aerosol pollution and the diurnal changes of heavy

rainfall and associated clouds in the BTH region. However, although this work has attempted to exclude the
impacts from the meteorological background particularly circulation and moisture, the observation study still
has its limitations on studying aerosol effects on rainfall and clouds: first, the noise and uncertainty of
different observational data cannot be avoid, e.g., the misdetection of CF in the satellite product when AOD is
large (Brennan et al., 2005; Levy et al., 2013) and the mutual interference between liquid and ice clouds (Holz
et al., 2008; Platnick et al., 2017); Second, the meteorological co-variations cannot be completely removed
thus bring the uncertainties, e.g., the meteorology might have a vital influence on the relationship of AOD and
CF (Quaas et al., 2010; Grandey et al., 2013) and the relationship of AOD and CTP (Gryspeerdt et al., 2014a);
Third, the different types of aerosols cannot be completely separated, although we used AI index and AOD of
BC/sulfate to distinguish the respective effects of absorbing aerosols and scattering aerosols. In addition, we
selected the extreme ranges of AOD/CDNC to compare the characteristics of heavy rainfall and associated
clouds, which could bring uncertainties that these extreme conditions might be related with totally different
microphysical process or meteorological background. So we further checked the results using the middle
range of AOD and CDNC such as $25^{th} - 50^{th}$ percentile versus $50^{th} -75^{th}$ percentile. The results are basically
the same and significant except the change of the peak time of heavy rainfall is not significant based on AOD.
The influence of aerosol pollution on the heavy rainfall and clouds may not be linear, but this study using the
idea of comparative experiments gives the observational results of the relationships between them. Numerical
model simulations are necessarily applied to further study the specific impact of aerosols on the heavy rainfall.
And the detailed processes of aerosol effect on the precipitation formation of mix-phased clouds also needs
further exploration in our future study.

**Data availability**

We are grateful to the National Meteorological Information Centre (NMIC) of the China Meteorological
Administration (CMA) for providing hourly precipitation datasets. MODIS aerosol and cloud data were
obtained from http://ladsweb.modaps.eosdis.nasa.gov; ultraviolet AI data from OMI was obtained from
https://daac.gsfc.nasa.gov/datasets?keywords=OMI&page=1; MACC-II and ERA-interim reanalysis datasets
were obtained from http://apps.ecmwf.int/datasets.

**Author contributions**

JY and SZ conceived the study. SZ processed data and drew the figures. SZ and JY analyzed the observational
results and WCW, CZ and DG gave the professional guidance. PS provided the hourly precipitation dataset.
SZ and JY prepared the manuscript with contributions from WCW and CZ.

**Competing interests**

The authors declare that they have no conflict of interest.

**Acknowledgements**

Jing Yang, Daoyi Gong & Peijun Shi are supported by funds from the National Natural Science Foundation of
China (41621061 and 41775071) and the National Key Research and Development Program-Global Change
and Mitigation Project: Global Change Risk of Population and Economic System: Mechanism and
Assessment (2016YFA0602401), Siyuan Zhou is supported by funds from State Key Laboratory of Earth
Surface Processes and Resource Ecology and Key Laboratory of Environmental Change and Natural Disaster.
Wei-Chyung Wang acknowledges the support of grants (to SUNYA) from the Office of Sciences (BER), U.S.
DOE and the U.S. National Science Foundation (1545917) in support of the Partnership for International
Research and Education project at the University at Albany. We deeply appreciate two anonymous referees
for their in-depth comments and constructive suggestions.

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

 **Tables**

| Indicator | Source | Begin time | Thresholds | |
|---|---|---|---|---|
| | | | 25th percentile | 75th percentile |
| AOD | MODIS | 2002 | 0.98 | 2.00 |
| CDNC ($cm^{-3}$) | MODIS | 2002 | 30.10 | 91.03 |
| AAI | OMI | 2005 | 0.13 | 0.52 |
| SAI | OMI | 2005 | - 0.13 | - 0.35 |
| AOD of BC | MACC | 2003 | 0.04 | 0.06 |
| AOD of sulfate | MACC | 2003 | 0.46 | 0.87 |
| SH at 850 hPa (g/kg) | ERA-interim | 2002 | 9.96 | 12.95 |


Table 1. The indicators of aerosols and moisture used in the study and their sources, begin times and the
thresholds (25th and 75th percentiles). The end time of all data is to 2012.




| Characteristics of heavy rainfall | Clean | | Polluted | | Difference | | Significance | |
|---|---|---|---|---|---|---|---|---|
| | AOD | CDNC | AOD | CDNC | AOD | CDNC | AOD | CDNC |
| Start time | 24.2 (3.9) | 24.3 (4.0) | 23.5 (4.8) | 22.9 (3.9) | - 0.7 | - 1.4 | P<0.05 | P<0.05 |
| Peak time | 23.0 (4.0) | 22.1 (5.3) | 22.0 (4.8) | 19.1 (5.7) | - 1.0 | - 3.0 | P<0.05 | P<0.05 |
| Duration | 4.0 (2.1) | 5.5 (3.3) | 4.8 (2.8) | 7.7 (4.3) | 0.8 | 2.2 | P<0.05 | P<0.05 |
| Intensity | 164.9 (98.4) | 166.0 (89.3) | 169.6 (94.3) | 162.7 (89.1) | 4.7 | - 3.3 | P>0.1 | P>0.1 |


Table 2. The mean values of start time (units: LST), peak time (units: LST), duration (units: hours) and
intensity (units: 0.1mm/hour) of heavy rainfall respectively on the clean and polluted conditions using two
indicators of AOD and CDNC, and their differences (polluted minus clean) and significances. The numbers in
the brackets stand for the standard deviations on the means. "P<0.05" stands for the difference has passed the
significance test of 95%, and "P>0.1" stands for the difference did not pass the significance test of 90%.


| Characteristics of heavy rainfall | AAI | SAI | Difference (AAI-SAI) | Less BC | More BC | Difference (More-Less) | Less sulfate | More sulfate | Difference (More-Less) |
|---|---|---|---|---|---|---|---|---|---|
| Start time | 23.4 (4.8) | 24.1 (4.4) | -0.7 | 24.2 (4.8) | 23.9 (4.4) | -0.3 | 24.0 (4.3) | 24.5 (4.4) | 0.5 |
| Peak time | 21.0 (5.3) | 22.6 (5.1) | -1.6 | 23.4 (5.3) | 22.3 (4.0) | -1.1 | 23.2 (4.5) | 22.9 (4.8) | -0.3 |
| Duration | 5.0 (3.1) | 6.0 (3.8) | -1.0 | 4.8 (2.6) | 4.6 (2.7) | -0.2 | 4.0 (2.1) | 5.5 (3.0) | 1.5 |

Table 3. The mean values of start time (units: LST), peak time (units: LST) and duration (units: hours) of heavy rainfall respectively on the conditions with more absorbing aerosols (AAI more than 75[th] percentile, from OMI), more scattering aerosols (SAI more than 75[th] percentile, from OMI), less or more BC (AOD of BC less than 25[th] or more than 75[th] percentile, from MACC), less or more sulfate (AOD of sulfate less than 25[th] or more than 75[th] percentile, from MACC), and their differences. Numbers in the brackets stand for the standard deviations on the means. All differences have passed the significant test of 95%.

| Clean/Polluted | | CF | CTP | COT | | CWP | | CER | |
|---|---|---|---|---|---|---|---|---|---|
| | | | | liquid | ice | liquid | ice | liquid | ice |
| AOD | Clean | 62.8 (17.6) | 442.3 (149.6) | 6.9 (4.5) | 6.7 (8.5) | 62.8 (36.6) | 123.1 (168.9) | 16.7 (4.4) | 32.0 (8.7) |
| | Polluted | 89.3 (12.9) | 487.3 (145.7) | 10.0 (5.8) | 12.9 (17.0) | 96.4 (52.5) | 211.3 (279.3) | 17.5 (3.5) | 29.2 (9.0) |
| CDNC | Clean | 94.5 (6.1) | 398.0 (131.7) | 8.1 (6.0) | 8.7 (10.6) | 102.4 (104.3) | 171.6 (204.3) | 20.4 (2.8) | 34.2 (6.0) |
| | Polluted | 97.4 (4.2) | 430.8 (135.2) | 40.4 (21.5) | 33.1 (22.7) | 318.2 (213.2) | 542.5 (447.8) | 12.2 (1.9) | 25.4 (8.7) |

Table 4. The mean values of CF (units: %), CTP (units: hPa), COT (liquid and ice, units: none), CWP (liquid and ice, units: g/m$^2$) and CER (liquid and ice, units: μm) from MODIS C6 cloud product on the clean condition (less than 25[th] percentile) and polluted condition (more than 75[th] percentile) using two indicators of AOD and CDNC. Numbers in the brackets stand for the standard deviations on the means. Numbers in grey indicate the results of liquid COT & CER are related to the calculation of CDNC. The differences between clean and polluted conditions have all passed the significant test of 95%.

| Group (case number) | | CF | CTP | COT | | CWP | | CER | |
|---|---|---|---|---|---|---|---|---|---|
| | | | | liquid | ice | liquid | ice | liquid | ice |
| 1 | Clean, dry (153) | 93.8 (6.1) | 393.3 (117.3) | 7.2 (4.6) | 7.6 (9.4) | 88.7 (70.6) | 149.0 (146.4) | 20.4 (3.0) | 36.7 (6.6) |
| 2 | Polluted, dry (128) | 95.6 (5.1) | 475.7 (142.8) | 50.2 (24.4) | 43.4 (19.3) | 424.6 (275.5) | 793.5 (404.7) | 12.6 (2.4) | 30.0 (7.0) |
| 3 | Clean, wet (155) | *92.7 (7.0)* $p_{1,3}>0.05$ | 457.4 (191.0) | 8.6 (4.7) | 10.6 (12.6) | 101.9 (64.5) | 207.7 (254.1) | *19.8 (2.5)* $p_{1,3}>0.05$ | 33.2 (4.4) |
| 4 | Polluted, wet (194) | 97.8 (4.4) | *419.7 (141.0)* $p_{3,4}>0.05$ | 36.4 (20.6) | 28.4 (21.1) | 295.9 (208.7) | 456.4 (412.1) | *12.5 (2.0)* $p_{2,4}>0.1$ | 24.4 (7.5) |

Table 5. The mean values of CF (units: %), CTP (units: hPa), COT (liquid and ice, units: none), CWP (liquid and ice, units: g/m$^2$) and CER (liquid and ice, units: μm) in four groups. Numbers in the brackets stand for the standard deviations on the means. Italic numbers in grey represent that the differences are not significant, in which "P>0.05" stands for the difference has passed the significance test of 90% but did not pass the significance test of 95%, and "P>0.1" stands for the difference did not pass the significance test of 90%.


## **Figures**


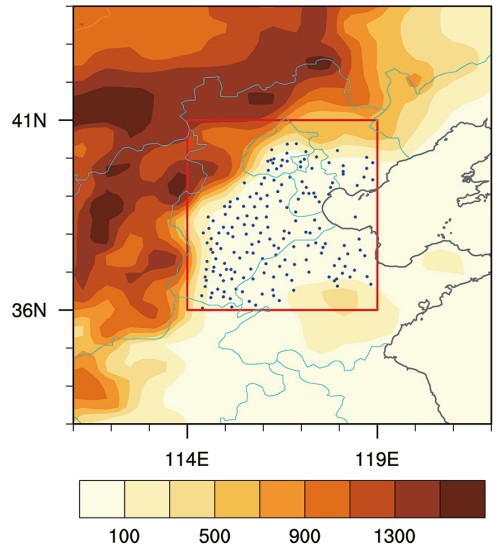


Figure 1. Selected rainfall stations (blue dots) and topography (shading, units: m) in the BTH region (red box,

36–41° N, 114–119° E).


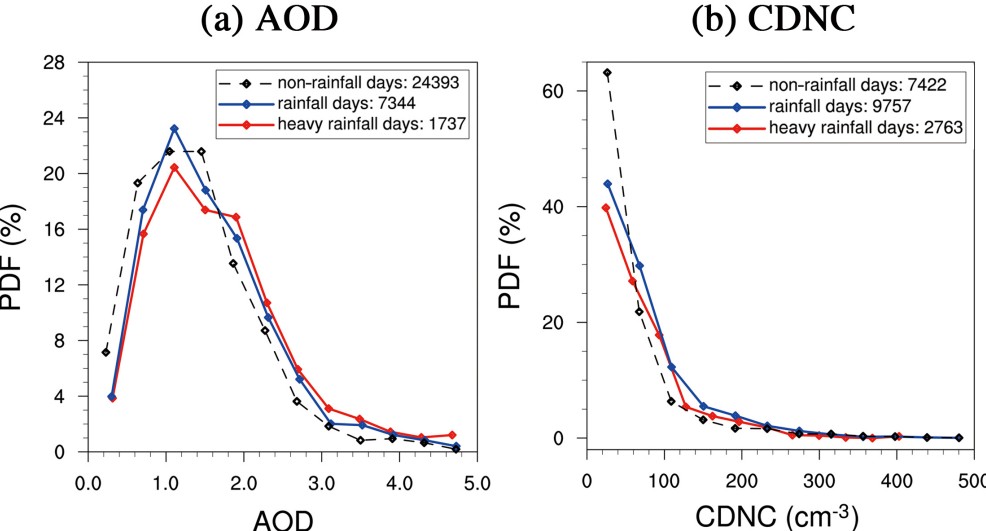


Figure 2. PDF of (a) AOD and (b) CDNC (cm$^{-3}$) (data from MODIS) on non-rainfall days (black lines),

rainfall days (blue lines) and heavy rainfall days (red lines) in southwesterly during early summers from 2002

to 2012. Numbers in the legends denote the sample number.



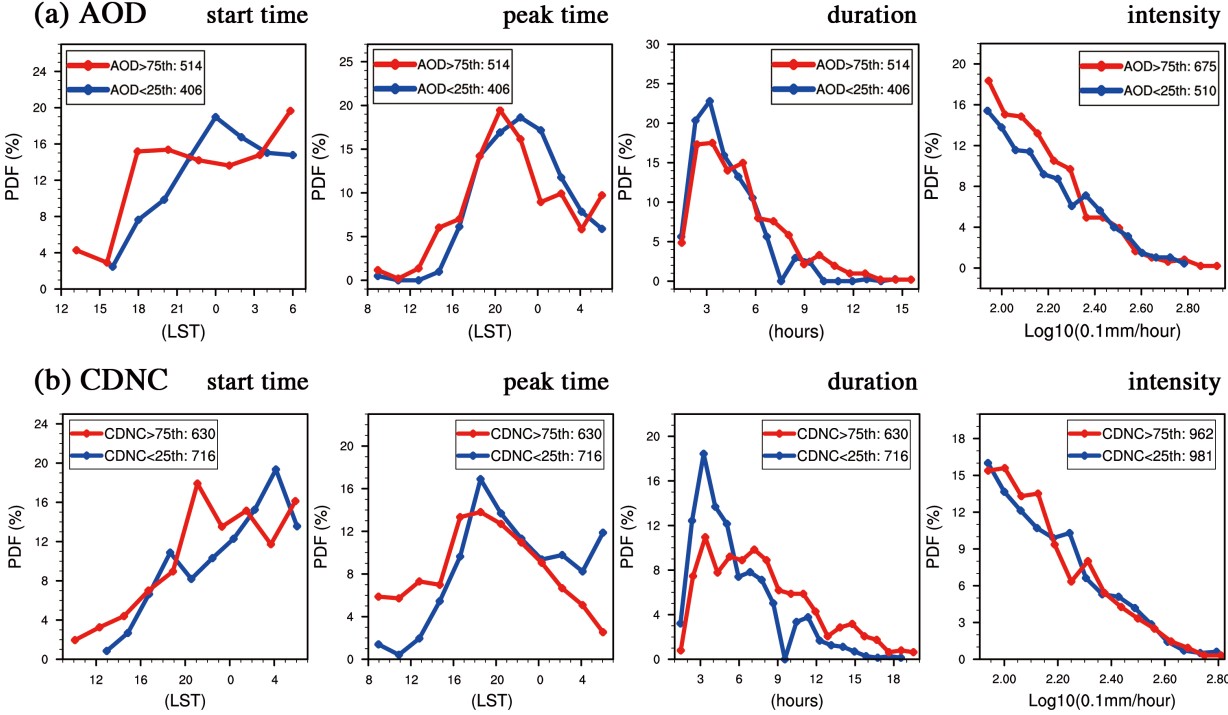

Figure 3. PDF of start time (units: LST), peak time (units: LST), duration (units: hours) and intensity (units:
0.1mm/hour) of heavy rainfall (data from CMA) on selected clean (blue lines) and polluted (red lines)
conditions, respectively using indicator of (a) AOD and (b) CDNC (cm$^{-3}$), during early summers from 2002 to

992 2012.


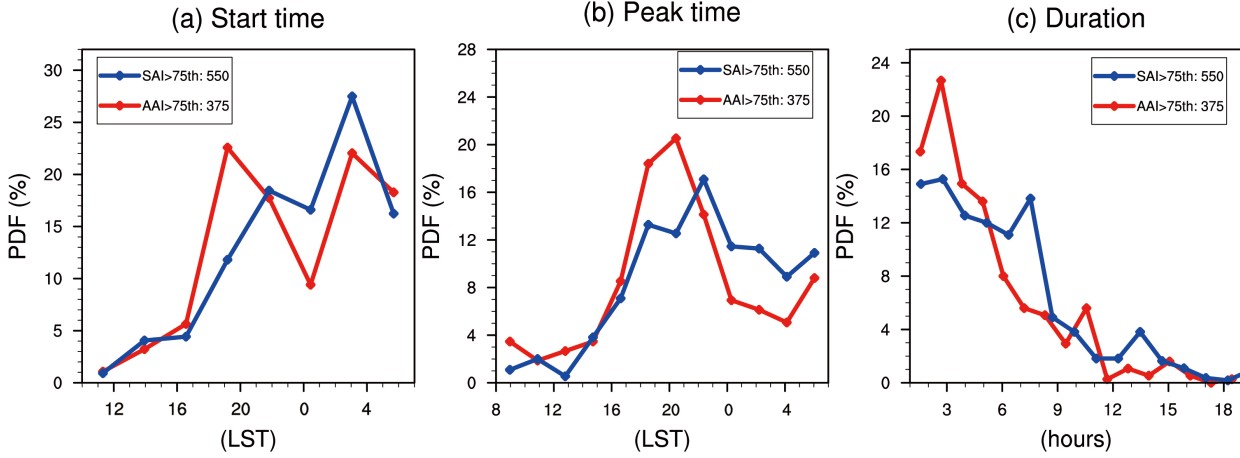


Figure 4. PDF of (a) start time (units: LST), (b) peak time (units: LST), and (c) duration (units: hours) of
heavy rainfall on the days with SAI more than 75[th] percentile (blue lines, data from OMI) and days with AAI
more than 75[th] percentile (red lines, data from OMI), during early summers from 2005 to 2012.


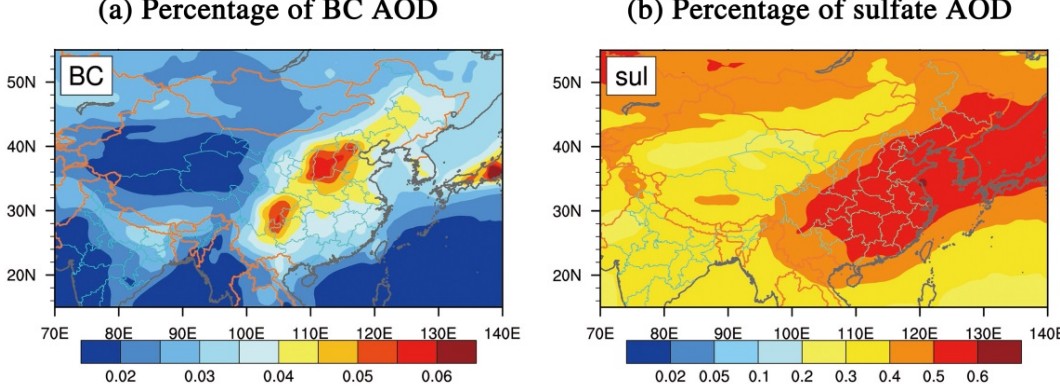

.000

.001 Figure 5. Percentages of AOD for (a) BC and (b) sulfate from MACC reanalysis data in summers (June –

.002 August) during 2002 to 2012.

.003

.004

.005

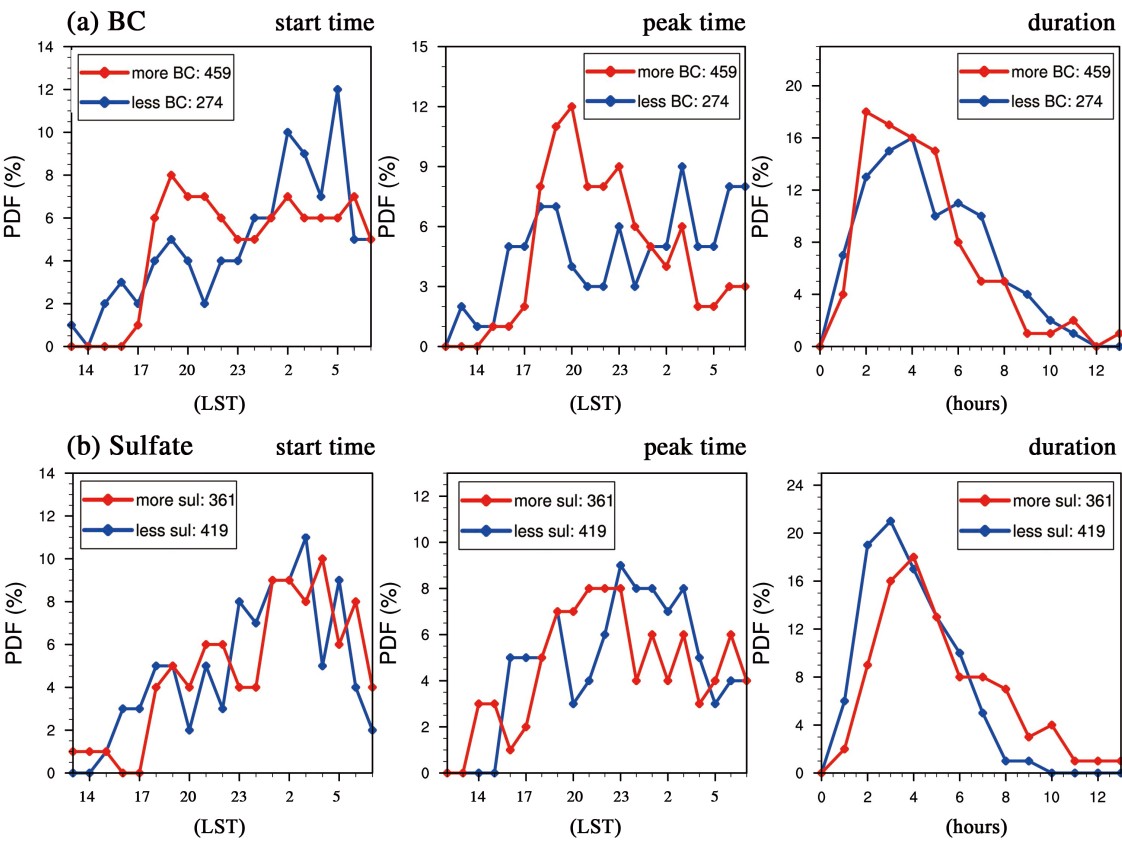

.006

.007 Figure 6. PDF of start time (units: LST), peak time (units: LST) and duration (units: hours) of heavy rainfall

.008 on the different conditions of (a) BC and (b) sulfate. Blue/red lines stand for the condition of less/more BC or

.009 sulfate (AOD of BC or sulfate less than 25[th] /more than 75[th] percentile, data from MACC) during early

.010 summers from 2003 to 2012.

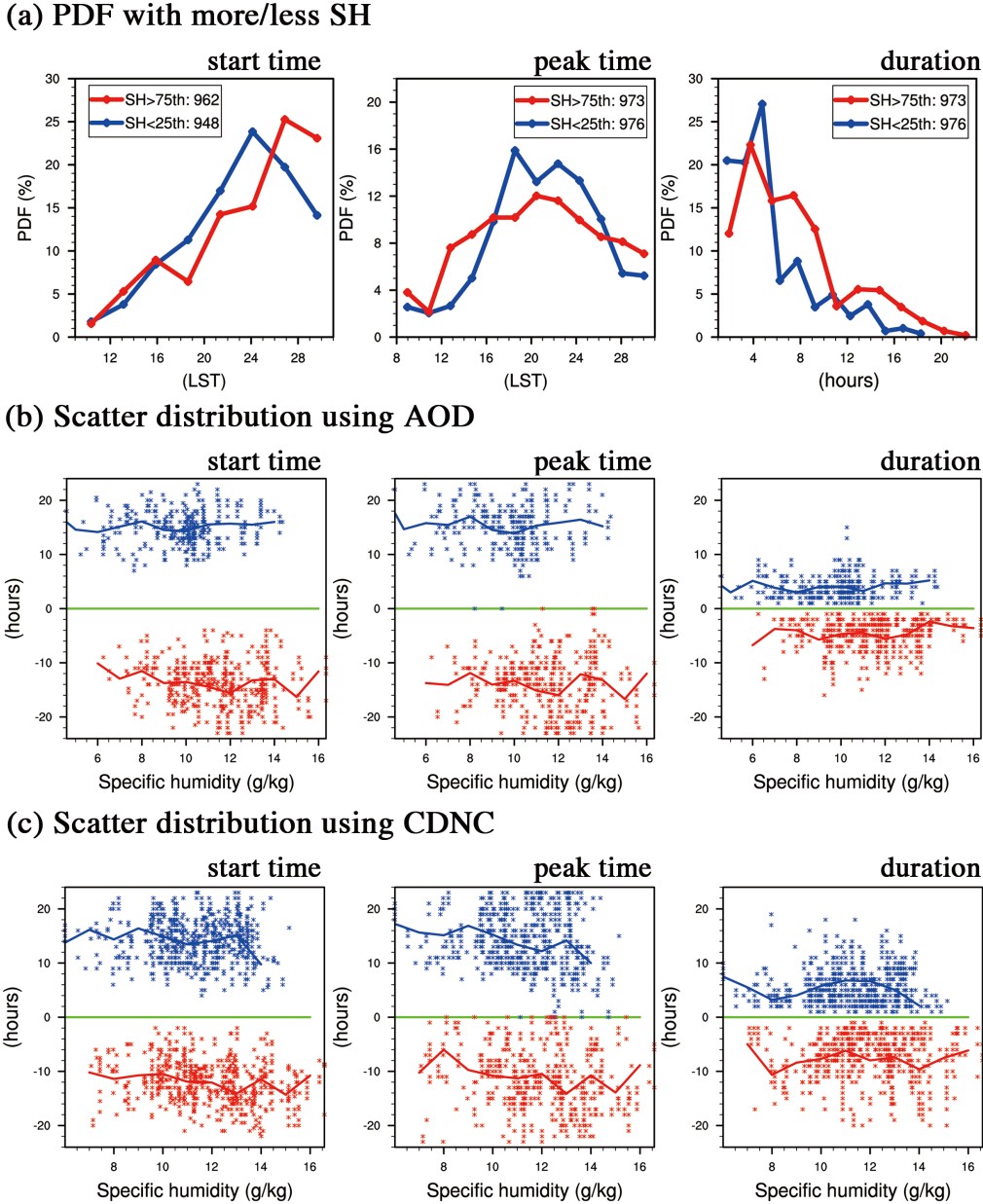

.014    Figure 7. (a) PDF of start time (units: LST), peak time (units: LST), and duration (units: hours) of heavy

.015    rainfall with less moisture (blue lines, SH at 850 hPa less than 25[th] percentile, data form ERA-interim) and

.016    more moisture (red lines, SH at 850 hPa more than 75[th] percentile, data form ERA-interim). (b) and (c) are

.017    scatter distributions of SH-start time/peak time/duration for clean cases (blue points) and polluted cases (red

.018    points) respectively using AOD and CDNC. Green lines stands for the start/peak time at 8:00 LST or the

.019    duration is 0 hours. Positive (negative) values stand for the hours away from 8:00 LST or 0 hours in clean

.020    (polluted) cases. Blue (red) lines stand for the mean values of rainfall characteristics at each integer of SH in

.021    clean (polluted) cases.

.022

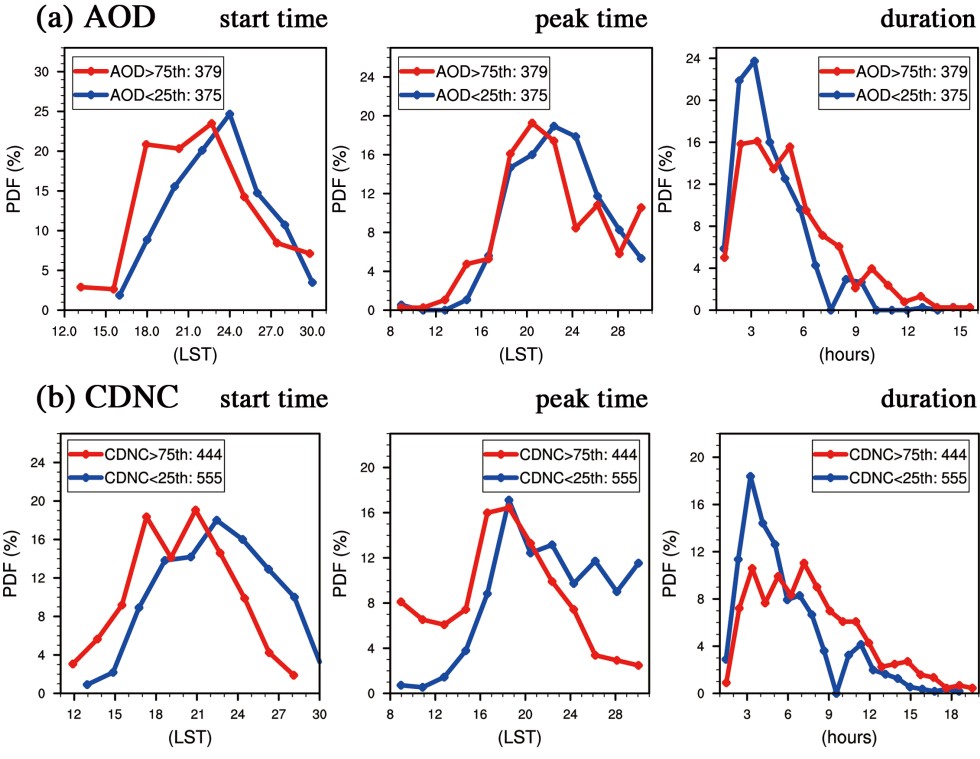

.026    Figure 8. PDF of start time (units: LST), peak time (units: LST), and duration (units: hours) of heavy rainfall

.027    on selected clean (blue lines) and polluted (red lines) conditions with SH at 850 hPa (from ERA-interim) less

.028    than 75th percentile, respectively using indicator of (a) AOD and (b) CDNC ($cm^{-3}$), during early summers from

.029    2002 to 2012.

.030

.031

.032

.033

.034

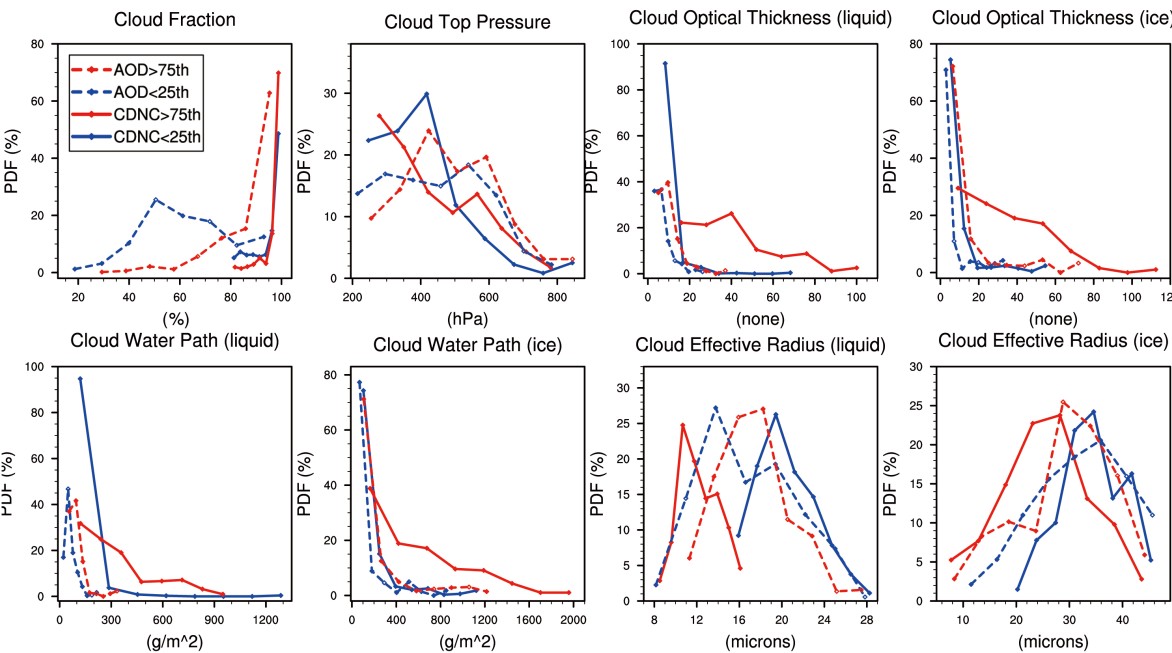

.035

.036    Figure 9. PDF of CF (units: %), CTP (units: hPa), COT (liquid and ice, units: none), CWP (liquid and ice,

.037    units: g/m$^2$) and CER (liquid and ice, units: μm) on selected clean (blue dash lines: AOD<25[th] percentile; blue

.038    solid lines: CDNC<25[th] percentile) and polluted (red dash lines: AOD>75[th] percentile; red solid lines:

.039    CDNC>75[th] percentile) heavy rainfall days. All cloud variables are obtained from MODIS C6 cloud product.

.040

.041

.042

.043

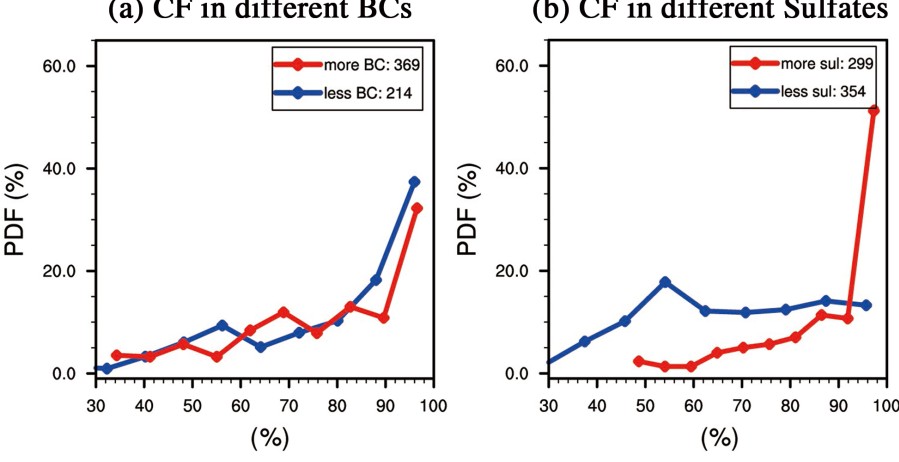

.044

.045    Figure 10. PDF of CF (units: %, data from MODIS) respectively for the conditions of less BC/sulfate (blue

.046    lines, AOD of BC/sulfate less than 25[th] percentile, data from MACC) and more BC/sulfate (red lines, AOD of

.047    BC/sulfate more than 75[th] percentile, data from MACC) cases with heavy rainfall during 10 early summers

.048    (2003-2012).

.052

Figure 11. A schematic diagram for aerosol impacts on heavy rainfall over Beijing-Tianjin-Hebei region.

.054

.055

.056