# Peer review of "An observational study of the effects of aerosols on diurnal variation of heavy rainfall and associated clouds over Beijing-Tianjin-Hebei Siyuan Zhou1,2,3, Jing Yang1,2\*, Wei-Chyung Wang3, Chuanfeng Zhao4, Daoyi Gong1,2, Peijun Shi1,2</sup"

_Atmospheric Chemistry and Physics, 2018_

## Referee Comment (RC1) · Anonymous Referee #1 · 1 Dec 2018

Major:

The authors attempt to empirically link rainfall and other cloud properties to AOD with the implied underlying mechanism that AOD~CCN at cloud base. The use of AOD as a proxy for CCN is not robust and has been discussed at length in the literature for many years (see Shinozuka et al. (2015) and references therein). This particular calculation is mainly problematic because AOD is increased by humidity (Twohy et al., 2009), which is also important to rainfall and the various cloud properties the authors are attempting to link to AOD. The authors show that RH covaries with AOD (Fig. 4a), but this is cast in the light of sources of moisture and pollution covarying, which

may be partially true, but ignores the leading order problem with their analysis. The authors are guaranteed a correlation between AOD and CF, rain rate, and so on, just by aerosol swelling and increasing scattering. It has been shown that when these effects are taken into account the correlation between AOD and cloud properties becomes much much weaker(Christensen et al., 2017). For the paper to be acceptable to be published the authors must use a different proxy for CCN that doesn't build in the result. One possibility would be for the authors to use something like cloud droplet number concentration (CDNC) instead of AOD. This is also a problematic technique as CDNC may potentially be affected by cloud structure/heterogeneity, which will be different between precipitating and non-precipitating clouds. The authors do use CER in their analysis, but this implicitly builds in variations in liquid water content. For example, in a cloud with a fixed CCN and thus a fixed CDNC the LWC can decrease and decrease the CER. The authors do use MACC reanalysis aerosol data as well. They could instead try and rely on the aerosol mass from this product, which has been shown to have skill in predicting variations in CDNC in previous studies(McCoy et al., 2018;Boucher and Lohmann, 1995;Lowenthal et al., 2004). However, in doing this they need to deal with whether variations in precipitation are driving aerosol.

Overall- the built-in correlation between RH and AOD via swelling and the inability to show causality makes this paper unpublishable in ACP.

The authors treat MERRA2 cloud properties like they are observations. They are not. MERRA2 does NOT ingest cloud properties (McCarty et al., 2016). MERRA2 creates clouds just like any other GCM (Molod et al., 2015) and the cloud fraction is not reflective of the observations (see Fig. 9 of Molod et al. (2015). For example- on L223 P7 the authors refer to the 3D cloud properties. I believe this is just the MERRA2 cloud properties based on the discussion on L147 P5.

The authors may want to consider consulting a copy editor as some of the statements are hard to parse as written. Because the paper must remove the AOD as a proxy of CCN and remove MERRA2 cloud properties I have not bothered to note all the places

that the English needs to be clarified.

The abstract is on the short side. This makes it hard to tell what they are actually doing in the paper. The description is very vague. Things like aerosol and humidity are discussed, but it is unclear precisely what they mean. Is it AOD, aerosol number in the PBL? Is it RH or specific humidity? Is it in the PBL or the free troposphere.

Specific comments

L65 P2: Complicacy is a word, but I am not sure it's fair to say that the complexity of the clouds alone leads to complexity in the indirect effects. Aerosol processes are also important.

L70 P3: The author refers to the Albrecht/lifetime/adjustments and the Twomey effect/first indirect effect as the Twomey effect.

L74 P3: What is the different condition of moisture? I think what the authors mean is that if the air near cloud is moist or not the cloud droplet size can increase or decrease. This sentence needs a bit of clarification. The Twomey effect is not related to cloud droplet radius. It is the relation between CCN and Nd.

L82 P3: In one day of what? Do the authors mean that convective rainfall usually starts and stops within a day?

L111 P4: I assume this means above sea level. Also, wouldn't the orographic effect be the lifting by the orography, so the slope and the advection, not just the height? Limiting to sub-100m is probably ok, but there needs to be a better discussion of this.

L114 P4: AOD is not a proxy for particle number. It is the brightness of aerosol in the column of atmosphere, which in turn is a function of the scattering of the particles, the number of particles, and the mass of particles. The authors want to relate this quantity to CCN, which is a very outdated idea. CCN is more directly related to AI, but even then this is a highly imperfect metric because it is sensitive to aerosol swelling by moisture and the vertical structure. See, for example, Shinozuka et al. (2015). The

use of AOD as a proxy for CCN in this paper is a critical error and makes it hard to draw any meaningful conclusions from their analysis because AOD is enhanced by RH (Christensen et al., 2017;Twohy et al., 2009).

L134 P4: The spatial resolution is not 1x1° in MACC. The authors presumably mean the data is regridded to 1x1°.

L141 P4: Provide a citation for the evaluation of MODIS CTH with CALIPSO.

L147 P5: The authors appear to be using cloud properties from MERRA2 as if they were observations. This is a model product and is not nudged to agree with observations in any way. This is another critical flaw in the paper.

L159 P5: The ECMWF-Interim resolution is not 1x1°- again I assume this is the gridded resolution used in the study, but is confused by saying MERRA2 is 0.624x0.5° (L152).

L184 P6: How was the t-test applied? Was it difference in means or difference in the means within a bin. Throughout the paper it is stated that results are significant at 95% confidence, but it is unclear what this actually means.

L223 P7: MERRA2 is NOT equivalent to observations of cloud properties. This is just an analysis of the modeled cloud properties in MERRA2.

L239 P8: Different moisture conditions (which I guess just means RH at a fixed pressure level) also affect clouds.

L240 P8: Fixing the wind direction is not enough to orthogonalize RH, AOD, and clouds.

L300 P9: The authors do try and use BC and SO4 mass concentrations, which sort of gets around the issues with AOD and RH. The model level that the concentration is taken at is not specified. However, the authors still partition by AOD, and just use BC and SO4 to partition the high and low AOD cases. The results shown in Fig. 8 are not terribly convincing. It is stated that the results are significant at 95% confidence. What does this mean? Are the means different and 95% confidence or are the values of the

PDF in each bin different at 95% confidence? From just looking at the PDFs it seems like only the BC peak time and SO4 duration are all that are different. The CF PDFs in Fig. 9 look different for SO4, but the jump in the PDF of CF at 90% does not look terribly robust. This may change if more bins are used in the PDF.

Boucher, O., and Lohmann, U.: The sulfate-CCN-cloud albedo effect, Tellus B, 47, 281-300, 10.1034/j.1600-0889.47.issue3.1.x, 1995. Christensen, M. W., Neubauer, D., Poulsen, C., Thomas, G., McGarragh, G., Povey, A. C., Proud, S., and Grainger, R. G.: Unveiling aerosol-cloud interactions Part 1: Cloud contamination in satellite products enhances the aerosol indirect forcing estimate, Atmos. Chem. Phys. Discuss., 2017, 1-21, 10.5194/acp-2017-450, 2017. Lowenthal, D. H., Borys, R. D., Choularton, T. W., Bower, K. N., Flynn, M. J., and Gallagher, M. W.: Parameterization of the cloud droplet-sulfate relationship, Atmos. Environ., 38, 287-292, 10.1016/j.atmosenv.2003.09.046, 2004. McCarty, W., Coy, L., R, G., A, H., Merkova, D., EB, S., M, S., and K, W.: MERRA-2 Input Observations: Summary and Assessment, Technical Report Series on Global Modeling and Data Assimilation, 46, 2016. McCoy, D. T., Bender, F. A. M., Grosvenor, D. P., Mohrmann, J. K., Hartmann, D. L., Wood, R., and Field, P. R.: Predicting decadal trends in cloud droplet number concentration using reanalysis and satellite data, Atmospheric Chemistry and Physics, 18, 2035-2047, 10.5194/acp-18-2035-2018, 2018. Molod, A., Takacs, L., Suarez, M., and Bacmeister, J.: Development of the GEOS-5 atmospheric general circulation model: evolution from MERRA to MERRA2, Geosci. Model Dev., 8, 1339-1356, 10.5194/gmd-8-1339-2015, 2015. Shinozuka, Y., Clarke, A. D., Nenes, A., Jefferson, A., Wood, R., McNaughton, C. S., Ström, J., Tunved, P., Redemann, J., Thornhill, K. L., Moore, R. H., Lathem, T. L., Lin, J. J., and Yoon, Y. J.: The relationship between cloud condensation nuclei (CCN) concentration and light extinction of dried particles: indications of underlying aerosol processes and implications for satellite-based CCN estimates, Atmos. Chem. Phys., 15, 7585-7604, 10.5194/acp-15-7585-2015, 2015. Twohy, C. H., Coakley, J. A., and Tahnk, W. R.: Effect of changes in relative humidity on aerosol scattering near clouds, Journal of Geophysical Research: Atmospheres, 114, n/a-n/a, 10.1029/2008JD010991, 2009.

---

## Referee Comment (RC2) · Anonymous Referee #2 · 4 Jan 2019

In this work, the authors look at how properties of gauge-measured rainfall is linked to satellite and reanalysis aerosol and cloud properties. They demonstrate that there are strong correlations between the satellite retrieved aerosol and cloud properties, similar to previous work. They also show a correlation between the timing of the precipitation and the retrieved aerosol. Using reanalysis meteorological and aerosol data, they demonstrate that these relationships between aerosol, cloud and precipitation vary as a function of large-scale humidity and aerosol type.

This work contains several interesting ideas, the use of the gauge precipitation data overcomes issues with the satellite retrieved datasets used in previous studies and the

authors have used their knowledge of the local meteorology to attempt to account for meteorological covariations. However, it is not clear that these covariations have actually been accounted for. Unfortunately, this means that many of the inferences in this work may be overstating the role of aerosols. The results in this paper are potentially interesting, but the authors either need to show more definitively that aerosols are driving these relationships or to tone down the assertions that aerosols are the controlling factor. As such, I would only recommend publication after major corrections.

**1 Main points**

**1.1 The role of aerosols**

Many previous studies have shown that aerosol optical depth (AOD) is not a good proxy for CCN (Stier, 2016) and is strongly correlated to humidity, which can generate correlations between AOD and cloud fraction (CF; Quaas et al., 2010), as well as other cloud properties (Christensen et al., 2017). It has been shown that using large scale humidity to account for this issue is insufficient (Boucher and Quaas, 2012).

This can even affect studies of cloud development similar to those in this study (e.g. Matsui et al., 2006; Meskhidze et al., 2009; Gryspeerdt et al., 2014). This is due to the AOD-CF relationship resulting in different initial cloud distributions for the high and low AOD populations. As cloud development is linked to this initial state, this leads to the AOD-CF relationship (known to be strongly controlled by relative humidity) generating a link between AOD and cloud (or precipitation) development.

As the authors note, it is not easy to isolate the impact of aerosols from that of meteorology in a purely observational study. By restricting the circulation patterns analysed, the authors have gone some way towards doing this, but subsetting by reanalysis humidity alone has been shown by previous studies to be unable to account for the impact

of meteorology. This is a difficult task, one that may not be achievable with current data. However, in that case, the conclusions would have to be changed to reflect this.

**1.2 Data choices**

The MERRA data is not suitable for use as a cloud product, as it is not a measurement, but a model parametrisation. I am not clear to the extent which MERRA represents aerosol-cloud interactions, but if they are not included in the model, the relationship between MERRA clouds (depending only on meteorology) and observed aerosol would be indicative of a meteorological covariation (similar to the results presented in Boucher and Quaas (2012)).

The MODIS 1 by 1 degree CTP is an average of multiple retrievals. In cases with multiple layers of cloud in the same gridbox, the average CTP may be less than 600hPa despite the gridbox containing large amounts of low cloud. The histograms in the MODIS product could be used to better ensure a low contribution of low level cloud.

To account for the impact of humidity on the AOD retrieval, it may be possible to use reanalysis aerosol (McCoy et al., 2017). However, care should be taken in strongly precipitating environments, as this has a quite different spatial sampling compared to MODIS. Whilst MACC/MERRA reanalysis may be able to account for uncertainties in the retrieval, it has quite different relationships to precipitation which should be considered if it is used (Gryspeerdt et al., 2015).

**1.3 Physical explanations**

I found some of the explanations of the aerosol-cloud correlations confusing. In particular, the explanation for the moisture dependence of the correlation between AOD and CTP (L278) is very different from the process described in the references.

The standard impact of aerosol on collision-coalesence (C-C) is to reduce its efficiency (as smaller droplets have a reduced coalesence probability) suppressing precipitation (Albrecht, 1989). However, the explanation in section 4.3 seems to be suggesting that increased aerosol and droplet numbers enhance C-C, making precipitation more likely. This is not supported by the references (as far as I understand them).

There may also be similar explanations, for example - an increase in the low cloud fraction with increasing low level humidity could increase the gridbox mean CTP. The already high cloud fraction in the high AOD cases might limit the impact of this meteorological covariation to the low AOD cases only, explaining the different relationships of humidity, AOD and CTP observed in Fig. 3f.

This is not to say that the authors explanation is wrong, but it should be better supported by references, or with calculations or data if it is a new hypothesis.

**2   Minor points**

L61 - Qian et al. - not in references

L65 - complicacy - complicated nature?

L69 - The Twomey effect is only the change in droplet number resulting in a change in cloud albedo, not the collective result of all aerosol effects on liquid clouds.

L82 - Gryspeerdt et al. (2014) showed a link between aerosol and precipitation development with another attempt to account for meteorological covariations.

L83 - Similar to this study, these other studies have shown aerosol is correlated with a change in precipitation, not that it causes the change.

L110 - A map of the stations/region used would be useful here

L114 - AOD is not necessarily a good proxy for CCN (e.g. Stier et al, 2016)

L125 - 'we suppose' - could this be checked?

L133 - assimilation definitely reduces the shortcomings of model simulations, but it is not clear that it 'overcomes' them completely.

L146 - QA of marginal or higher. Marginal is the lowest retrieval confidence other than 'no confidence'. Why is this choice made - does using a higher confidence level strongly impact the results?

L159 - Why is a different reanalysis used for the clouds and the meteorology?

L167 - I am not clear why focusing on this time period better identifies the effect of aerosol

L174 - These are very high values of AOD. Brennan et al. (2005) suggested that at AOD>0.6, aerosol is likely to be misclassified as cloud. Might this affect the results here?

L190 - There also appears to be a later peak in the precipitation rate at high AOD. Is it clear whether the peak has move earlier or later.

L206 - Given that much of the paper is about the development of precipitation, it might be good to point out that the cloud properties are measured at the same time as the aerosol. This is stated in the methods section, but a small reminder would be useful.

L260 - It is not clear what 'nearly 350hPa' means. 340 or 360hPa?

L287 - Twomey not Towmey

L290 - via enhanced collision-coalesence - this is not the mechanism stated in Yuan et al. 2008, where the positive AOD-effective radius relationship is related to changes in aerosol properties.

L295 - The Wegner-Bergeron-Findeisen process can act whenever supercooled liquid

and ice crystals co-exist. As long as liquid droplets exist, it should not depend directly on the supersaturation over liquid, although if the region is supersaturated with respect to liquid, the liquid droplets can also continue to grow.

L319 - These changes would shift the PDF of CF, but I am not sure it can be said that BC 'corresponds to a slight decrease of CF when CF is more than 90%' as the CF in an aerosol-free atmosphere is not know. Instead, it would be more accurate to use phrases such as 'cloud fractions larger than 90% are less common in high BC environments'.

**References**

Albrecht, B. A.: Aerosols, Cloud Microphysics, and Fractional Cloudiness, Science, 245, 1227–1230, https://doi.org/10.1126/science.245.4923.1227, 1989.

Boucher, O. and Quaas, J.: Water vapour affects both rain and aerosol optical depth, Nat. Geosci., 6, 4–5, https://doi.org/10.1038/ngeo1692, 2012.

Brennan, J., Kaufman, Y., Koren, I., and Rong Rong, L.: Aerosol-cloud interaction-Misclassification of MODIS clouds in heavy aerosol, IEEE T. Geosci. Remote, 43, 911–915, https://doi.org/10.1109/TGRS.2005.844662, 2005.

Christensen, M. W., Neubauer, D., Poulsen, C. A., Thomas, G. E., McGarragh, G. R., Povey, A. C., Proud, S. R., and Grainger, R. G.: Unveiling aerosol–cloud interactions – Part 1: Cloud contamination in satellite products enhances the aerosol indirect forcing estimate, Atmos. Chem. Phys., 17, 13 151–13 164, https://doi.org/10.5194/acp-17-13151-2017, 2017.

Gryspeerdt, E., Stier, P., and Partridge, D. G.: Links between satellite-retrieved aerosol and precipitation, Atmos. Chem. Phys., 14, 9677–9694, https://doi.org/10.5194/acp-14-9677-2014, 2014.

Gryspeerdt, E., Stier, P., White, B. A., and Kipling, Z.: Wet scavenging limits the detection of aerosol effects on precipitation, Atmos. Chem. Phys., 15, 7557–7570, https://doi.org/10.5194/acp-15-7557-2015, 2015.

Matsui, T., Masunaga, H., Kreidenweis, S. M., Pielke, R. A., Tao, W.-K., Chin, M., and Kaufman, Y. J.: Satellite-based assessment of marine low cloud variability associated with aerosol,

atmospheric stability, and the diurnal cycle, J. Geophys. Res., 111, 17 204, https://doi.org/
10.1029/2005JD006097, 2006.

McCoy, D. T., Bender, F. A.-M., Mohrmann, J. K. C., Hartmann, D. L., Wood, R., and Grosvenor,
D. P.: The global aerosol-cloud first indirect effect estimated using MODIS, MERRA, and
AeroCom, J. Geophys. Res., 122, 1779–1796, https://doi.org/10.1002/2016JD026141, 2017.

Meskhidze, N., Remer, L. A., Platnick, S., Negrón Juárez, R., Lichtenberger, A. M., and Aiyyer,
A. R.: Exploring the differences in cloud properties observed by the Terra and Aqua MODIS
Sensors, Atmos. Chem. Phys., 9, 3461–3475, https://doi.org/10.5194/acp-9-3461-2009, 2009.

Quaas, J., Stevens, B., Stier, P., and Lohmann, U.: Interpreting the cloud cover – aerosol
optical depth relationship found in satellite data using a general circulation model, Atmos.
Chem. Phys., 10, 6129–6135, https://doi.org/10.5194/acp-10-6129-2010, 2010.

Stier, P.: Limitations of passive remote sensing to constrain global cloud condensation nuclei,
Atmos. Chem. Phys., 16, 6595–6607, https://doi.org/10.5194/acp-16-6595-2016, 2016.

---

## Author Comment (AC1) · 22 Feb 2019

**Answers to Referee #1's comments**

**Major:**

The authors attempt to empirically link rainfall and other cloud properties to AOD with the implied underlying mechanism that AOD_CCN at cloud base. The use of AOD as a proxy for CCN is not robust and has been discussed at length in the literature for many years (see Shinozuka et al. (2015) and references therein). This particular calculation is mainly problematic because AOD is increased by humidity (Twohy et al., 2009), which is also important to rainfall and the various cloud properties the authors are attempting to link to AOD. The authors show that RH covaries with AOD (Fig. 4a), but this is cast in the light of sources of moisture and pollution covarying, which may be partially true, but ignores the leading order problem with their analysis. The authors are guaranteed a correlation between AOD and CF, rain rate, and so on, just by aerosol swelling and increasing scattering. It has been shown that when these effects are taken into account the correlation between AOD and cloud properties becomes much weaker (Christensen et al., 2017). For the paper to be acceptable to be published the authors must use a different proxy for CCN that doesn't build in the result. One possibility would be for the authors to use something like cloud droplet number concentration (CDNC) instead of AOD. This is also a problematic technique as CDNC may potentially be affected by cloud structure/heterogeneity, which will be different between precipitating and non-precipitating clouds. The authors do use CER in their analysis, but this implicitly builds in variations in liquid water content. For example, in a cloud with a fixed CCN and thus a fixed CDNC the LWC can decrease and decrease the CER. The authors do use MACC reanalysis aerosol data as well. They could instead try and rely on the aerosol mass from this product, which has been shown to have skill in predicting variations in CDNC in previous studies (McCoy et al., 2018;Boucher and Lohmann, 1995;Lowenthal et al., 2004). However, in doing this they need to deal with whether variations in precipitation are driving aerosol.

Overall- the built-in correlation between RH and AOD via swelling and the inability to show causality makes this paper unpublishable in ACP.

The authors treat MERRA2 cloud properties like they are observations. They are not. MERRA2 does NOT ingest cloud properties (McCarty et al., 2016). MERRA2 creates clouds just like any other GCM (Molod et al., 2015) and the cloud fraction is not reflective of the observations (see Fig. 9 of Molod et al. (2015). For example- on L223 P7 the authors refer to the 3D cloud properties. I believe this is just the MERRA2 cloud properties based on the discussion on L147 P5.

The authors may want to consider consulting a copy editor as some of the statements are hard to parse as written. Because the paper must remove the AOD as a proxy of CCN and remove MERRA2 cloud properties I have not bothered to note all the places that the English needs to be clarified.

The abstract is on the short side. This makes it hard to tell what they are actually doing in the paper. The description is very vague. Things like aerosol and humidity are discussed, but it is unclear precisely what they mean. Is it AOD, aerosol number in the PBL? Is it RH or specific humidity? Is it in the PBL or the free troposphere.

R: To respond fully the comments, the manuscript is completely re-written. The revised manuscript is attached and major changes are briefly summarized here:

1. We agree that the use of AOD has inherent issues as the reviewer pointed out. So in addition to AOD, two indicators, the retrieved CDNC and ultraviolet AI, are also selected in the analysis. CDNC calculated by COT and CER from MODIS, is used to separate the different CCN conditions and verify the results based on AOD; see Section 2.1.3. AI from OMI is used to identify rainfall days having absorbing aerosols versus scattering aerosols; see Section 2.1.2. The uncertainties associated with these indicators are given in Section 6.3.

2. For cloud properties, we use CDNC for CCN and the absolute (instead of relative) humidity for moisture. In addition, the analysis is conducted by dividing the heavy rainfall (larger than 8mm/hour) days into four groups to compare the effects of CCN and moisture individually on the cloud properties; see Section 5.2.

3. The results using MERRA2 reanalysis data are deleted.

4. A section of data interpolation is added; see Section 2.2.1.

**Specific comments**

1.L65 P2: Complicacy is a word, but I am not sure it's fair to say that the complexity of the clouds alone leads to complexity in the indirect effects. Aerosol processes are also important.

R: The statement is revised; see Lines 70-72, Page 3.

2. L70 P3: The author refers to the Albrecht/lifetime/adjustments and the Twomey effect/ first indirect effect as the Twomey effect.

R: The statement is corrected; see Lines 73-75, Page 3.

3. L74 P3: What is the different condition of moisture? I think what the authors mean is that if the air near cloud is moist or not the cloud droplet size can increase or decrease. This sentence needs a bit of clarification. The Twomey effect is not related to cloud droplet radius. It is the relation between CCN and Nd.

R: The different condition of moisture was referred to different moisture supply near the cloud base. In any case, we have re-written the sentence for clarity; see Lines 80-85, Page 3.

4. L82 P3: In one day of what? Do the authors mean that convective rainfall usually starts and stops within a day?

R: "In one day" refers to that the heavy convective rainfall over BTH region usually occurs within 24 hours, which is no more than 20 hours as shown in Fig.2 in the manuscript. The average duration of all heavy rainfall events is 6.1 hours. We modified the sentence for clarity; see Lines 88-90, Page 3.

5. L111 P4: I assume this means above sea level. Also, wouldn't the orographic effect be the lifting by the orography, so the slope and the advection, not just the height? Limiting to sub-100m is probably ok, but there needs to be a better discussion of this.

R: Thanks for your suggestion. The "below the topography of 100 meter" does mean above sea level. We just want to focus on the aerosol impact on the rainfall diurnal variation in the plain area in order to reduce the probable influence of orography lifting effect like you said on the rainfall diurnal variation as far as possible. Below 100 meters above the sea level is a simple criterion for choosing plain area, and in the study we temporarily don't consider other orography effects. We added a figure of orography and stations over the study region to show that we select the plain area; see Lines 123-126, Page 4.

6. L114 P4: AOD is not a proxy for particle number. It is the brightness of aerosol in the column of atmosphere, which in turn is a function of the scattering of the particles, the number of particles, and the mass of particles. The authors want to relate this quantity to CCN, which is a very outdated idea. CCN is more directly related to AI, but even then this is a highly imperfect metric because it is sensitive to aerosol swelling by moisture and the vertical structure. See, for example, Shinozuka et al. (2015). The use of AOD as a proxy for CCN in this paper is a critical error and makes it hard to draw any meaningful conclusions from their analysis because AOD is enhanced by RH (Christensen et al., 2017;Twohy et al., 2009).

R: Yes, we agree, CDNC is also used as another indicator.

7. L134 P4: The spatial resolution is not 1°x1° in MACC. The authors presumably mean the data is regridded to 1°x1°.

R: The MACC reanalysis data we downloaded is the version of retrieve NetCDF with spatial resolution of 1°x1° which is the same as the resolution of MODIS (the resolution of ECMWF data is optional, see http://apps.ecmwf.int/datasets/data/macc-reanalysis/levtype=sfc). To unify the datasets, we interpolated the gridded datasets onto stations using the average value in a 1°×1° grid as the background condition of each rainfall station; see Section 2.2.1.

8. L141 P4: Provide a citation for the evaluation of MODIS CTH with CALIPSO.

R: References are added; see Lines 159, Page 5.

9. L147 P5: The authors appear to be using cloud properties from MERRA2 as if they were observations. This is a model product and is not nudged to agree with observations in any way. This is another critical flaw in the paper.

R:We agree. The results using MERRA2 data are deleted.

10. L159 P5: The ECMWF-Interim resolution is not 1x1_- again I assume this is the gridded resolution used in the study, but is confused by saying MERRA2 is 0.624x0.5_ (L152).

R: Same with MACC reanalysis data, ERA-interim data we downloaded is also the version of retrieve NetCDF with spatial resolution of 1°x1°. The horizontal resolution of MERRA2 data we downloaded is 0.624°x0.5° (now we deleted the part of MERRA2), and then we interpolated all the gridded datasets onto rainfall stations using the average value in a 1°×1° grid as the background condition of each rainfall station; see Section 2.2.1.

11. L184 P6: How was the t-test applied? Was it difference in means or difference in the means within a bin. Throughout the paper it is stated that results are significant at 95% confidence, but it is unclear what this actually means.

R: Student's t-test is used to check the significance of differences in means between different groups not within a bin. And the numbers of bins in the study have been all tested for better representing the PDF distribution. We have re-written the sentence for clarity; see Lines 235-236, Page 7.

12. L223 P7: MERRA2 is NOT equivalent to observations of cloud properties. This is just an analysis of the modeled cloud properties in MERRA2.

R:We agree, this part is deleted.

13. L239 P8: Different moisture conditions (which I guess just means RH at a fixed pressure level) also affect clouds.

R: Yes, they are more explicit now; see Lines 331-332, Page 10.

14. L240 P8: Fixing the wind direction is not enough to orthogonalize RH, AOD, and clouds.

R: We perceive that the circulation, radiation, moisture condition and cloud all influence rainfall. However, changes of radiation and cloud may associate more directly with aerosols. We fix the wind direction to remove circulation influence. To investigate the influence of moisture supply on aerosol cloud effect, we have additionally divide the heavy rainfall days into four groups and compare the influences of CCN and moisture individually on the cloud properties; see Section 5.2. Nevertheless, the sentence is modified; see Lines 332-333 of Page 10.

15. L300 P9: The authors do try and use BC and $SO_4$ mass concentrations, which sort of gets around the issues with AOD and RH. The model level that the concentration is taken at is not specified. However, the authors still partition by AOD, and just use BC and $SO_4$ to partition the high and low AOD cases. The results shown in Fig. 8 are not terribly convincing. It is stated that the results are significant at 95% confidence. What does this mean? Are the means different and 95% confidence or are the values of the

PDF in each bin different at 95% confidence? From just looking at the PDFs it seems like only the BC peak time and SO4 duration are all that are different. The CF PDFs in Fig. 9 look different for SO4, but the jump in the PDF of CF at 90% does not look terribly robust. This may change if more bins are used in the PDF.

R: To respond to the comments, the ultraviolet AI from OMI is also used, which further confirm the findings. The significance refers to the significance of difference in means rather than in each bin. As shown in Fig. A1 below, the results are consistent for different bins. A statement is therefore added; see Lines 234-235 of Page 7.

[Figure]

Figure A1. PDF of CF (units: %) in different conditions of sulfate using (a) 10 bins, (b) 25 bins and (c) 40 bins. Blue/red lines stand for the condition of less/more sulfate.

Boucher, O., and Lohmann, U.: The sulfate-CCN-cloud albedo effect, Tellus B, 47, 281-300, 10.1034/j.1600-0889.47.issue3.1.x, 1995.

Christensen, M. W., Neubauer, D., Poulsen, C., Thomas, G., McGarragh, G., Povey, A. C., Proud, S., and Grainger, R. G.: Unveiling aerosol-cloud interactions Part 1: Cloud contamination in satellite products enhances the aerosol indirect forcing estimate, Atmos. Chem. Phys. Discuss., 2017, 1-21, 10.5194/acp-2017-450, 2017.

Lowenthal, D. H., Borys, R. D., Choularton, T. W., Bower, K. N., Flynn, M. J., and Gallagher, M. W.: Parameterization of the cloud dropletsulfate relationship, Atmos. Environ., 38, 287-292, 10.1016/j.atmosenv.2003.09.046, 2004.

McCarty, W., Coy, L., R, G., A, H., Merkova, D., EB, S., M, S., and K, W.: MERRA-2 Input Observations: Summary and Assessment, Technical Report Series on Global Modeling and Data Assimilation, 46, 2016.

McCoy, D. T., Bender, F. A. M., Grosvenor, D. P., Mohrmann, J. K., Hartmann, D. L., Wood, R., and Field, P. R.: Predicting decadal trends in cloud droplet number concentration using reanalysis and satellite data, Atmospheric Chemistry and Physics, 18, 2035-2047, 10.5194/acp-18-2035-2018, 2018.

Molod, A., Takacs, L., Suarez, M., and Bacmeister, J.: Development of the GEOS-5 atmospheric general circulation model: evolution from MERRA to MERRA2, Geosci. Model Dev., 8, 1339-1356, 10.5194/gmd-8-1339-2015, 2015.

[revised manuscript text omitted]

---

## Author Comment (AC2) · 22 Feb 2019

**Answers to Referee #2's comments**

In this work, the authors look at how properties of gauge-measured rainfall is linked to satellite and reanalysis aerosol and cloud properties. They demonstrate that there are strong correlations between the satellite retrieved aerosol and cloud properties, similar to previous work. They also show a correlation between the timing of the precipitation and the retrieved aerosol. Using reanalysis meteorological and aerosol data, they demonstrate that these relationships between aerosol, cloud and precipitation vary as a function of large-scale humidity and aerosol type.

This work contains several interesting ideas, the use of the gauge precipitation data overcomes issues with the satellite retrieved datasets used in previous studies and the authors have used their knowledge of the local meteorology to attempt to account for meteorological covariations. However, it is not clear that these covariations have actually been accounted for. Unfortunately, this means that many of the inferences in this work may be overstating the role of aerosols. The results in this paper are potentially interesting, but the authors either need to show more definitively that aerosols are driving these relationships or to tone down the assertions that aerosols are the controlling factor. As such, I would only recommend publication after major corrections.

Main points

1.1 The role of aerosols

Many previous studies have shown that aerosol optical depth (AOD) is not a good proxy for CCN (Stier, 2016) and is strongly correlated to humidity, which can generate correlations between AOD and cloud fraction (CF; Quaas et al., 2010), as well as other cloud properties (Christensen et al., 2017). It has been shown that using large scale humidity to account for this issue is insufficient (Boucher and Quaas, 2012).

This can even affect studies of cloud development similar to those in this study (e.g. Matsui et al., 2006; Meskhidze et al., 2009; Gryspeerdt et al., 2014). This is due to the AOD-CF relationship resulting in different initial cloud distributions for the high and low AOD populations. As cloud development is linked to this initial state, this leads to the AOD-CF relationship (known to be strongly controlled by relative humidity) generating a link between AOD and cloud (or precipitation) development.

As the authors note, it is not easy to isolate the impact of aerosols from that of meteorology in a purely observational study. By restricting the circulation patterns analysed, the authors have gone some way towards doing this, but subsetting by reanalysis humidity alone has been shown by previous studies to be unable to account for the impact of meteorology. This is a difficult task, one that may not be achievable with current data. However, in that case, the conclusions would have to be changed to reflect this.

R: We certainly are in agreement with the issues raised by the reviewer and the manuscript has been completely re-written. To respond the comments, we have done the following to address them. In addition to AOD, two indicators, the retrieved CDNC and ultraviolet AI, are also used in the analysis. CDNC calculated by COT and CER from MODIS, is used to separate the different CCN conditions and verify the results based on AOD; see Section 2.1.3. AI from OMI is used to identify rainfall days having absorbing aerosols versus scattering aerosols; see Section 2.1.2. The uncertainties associated with these indicators are given in Section 6.3.

For moisture effect on cloud properties, we use CDNC for CCN and the absolute (instead of relative) humidity for moisture. In addition, the analysis is conducted by dividing the heavy rainfall days into four groups to compare the effects of CCN and moisture individually on the cloud properties; see Section 5.2.

1.2 Data choices

The MERRA data is not suitable for use as a cloud product, as it is not a measurement, but a model parametrisation. I am not clear to the extent which MERRA represents aerosol-cloud interactions, but if they are not included in the model, the relationship between MERRA clouds (depending only on meteorology) and observed aerosol would be indicative of a meteorological covariation (similar to the results presented in Boucher and Quaas (2012)).

The MODIS 1 by 1 degree CTP is an average of multiple retrievals. In cases with multiple layers of cloud in the same gridbox, the average CTP may be less than 600hPa despite the gridbox containing large amounts of low cloud. The histograms in the MODIS product could be used to better ensure a low contribution of low level cloud.

To account for the impact of humidity on the AOD retrieval, it may be possible to use reanalysis aerosol (McCoy et al., 2017). However, care should be taken in strongly precipitating environments, as this has a quite different spatial sampling compared to MODIS. Whilst MACC/MERRA reanalysis may be able to account for uncertainties in the retrieval, it has quite different relationships to precipitation which should be considered if it is used (Gryspeerdt et al., 2015).

R:We agree on the remarks on MERRA's cloud data, and since it is a very minor part of the study, we have eliminated its use completely.

We regard the clouds with the CTP above 600hPa as the mixed-phase clouds, but actually the results of liquid cloud properties in the study (Figure 6 in the revised manuscript) might come from the liquid clouds (low-level clouds) and also the liquid part of mixed phase clouds since we cannot distinguish the liquid part of mixed-phase clouds from the pure liquid clouds in our observation study. We also checked the changes of pure liquid clouds (when the ice properties are missing) as shown in Figure A1, the results are the same as the clouds with CTP above 600hPa. We added sentences to clarify this issue; see Lines 332-336, Page 10.

On AOD data, both MACC and MERRA datasets show very similar features with MODIS (such as the result of MACC in Fig. A2). Since the MACC/MERRA reanalysis data are not completely equal to observational data, we did not address it in the revised manuscript.

[Figure]

Figure A1. PDF of CF (units: %), COT (units: none), CWP (units: g/m$^2$) and CER (units: μm) for only liquid clouds (when ice COT/CWP/CER is missing) on selected clean (blue lines: CDNC<25$^{th}$ percentile) and polluted (red lines: CDNC>75$^{th}$ percentile) heavy rainfall days.

[Figure]

Figure A2. PDF of (a) start time (units: LST), (b) peak time (units: LST), and (c) duration (units: hours) of heavy rainfall on selected clean (blue lines) and polluted (red lines) conditions using AOD from MACC.

1.3 Physical explanations

I found some of the explanations of the aerosol-cloud correlations confusing. In particular, the explanation for the moisture dependence of the correlation between AOD and CTP (L278) is very different from the process described in the references.

The standard impact of aerosol on collision-coalesence (C-C) is to reduce its efficiency (as smaller droplets have a reduced coalesence probability) suppressing precipitation (Albrecht, 1989). However, the explanation in section 4.3 seems to be suggesting that increased aerosol and droplet numbers enhance C-C, making precipitation more likely.

This is not supported by the references (as far as I understand them). There may also be similar explanations, for example - an increase in the low cloud fraction with increasing low level humidity could increase the gridbox mean CTP. The already high cloud fraction in the high AOD cases might limit the impact of this meteorological covariation to the low AOD cases only, explaining the different relationships of humidity, AOD and CTP observed in Fig. 3f.

This is not to say that the authors explanation is wrong, but it should be better supported by references, or with calculations or data if it is a new hypothesis.

R: As discussed in Section 6 in the revised manuscript, the cloud characteristics are much more complex and sensitive to the indicators used. While both AOD and CDNC indicate weaker invigoration effect, the liquid CER shows positive association with AOD but negative with CDNC. Therefore, in the revised manuscript, we only lay out the hypotheses and more insights will have to rely on model simulations.

Minor points

L61 - Qian et al. - not in references

R: Added.

L65 - complicacy - complicated nature?

R: We have modified it; see Lines 70-72, Page 3.

L69 - The Twomey effect is only the change in droplet number resulting in a change in cloud albedo, not the collective result of all aerosol effects on liquid clouds.

R: Thanks for pointing it out, we have revised it; see Lines 73-75, Page 3.

L82 - Gryspeerdt et al. (2014) showed a link between aerosol and precipitation development with another attempt to account for meteorological covariations.

R: Reference added.

L83 - Similar to this study, these other studies have shown aerosol is correlated with a change in precipitation, not that it causes the change.

R: We agree and modify the writing; see Lines 90-92, Page 3.

L110 – A map of the stations/region used would be useful here

R: Yes, added.

L114 – AOD is not necessarily a good proxy for CCN (e.g. Stier et al, 2016)

R: We agree and added CDNC as another proxy.

L125 - 'we suppose' - could this be checked?

R: As seen from the PDF of heavy rainfall start time in Fig. 2, all the events occur after 10:30 LST which is the overpass time of satellite, i.e., the AOD record we used is either before the precipitation starting in that rainfall day or on the previous day before the rainfall event. We have revised the writing; see Lines 133-136, Page 4.

L133 - assimilation definitely reduces the shortcomings of model simulations, but it is not clear that it 'overcomes' them completely.

R: The sentence is modified; see Line 149, Page 5.

L146 - QA of marginal or higher. Marginal is the lowest retrieval confidence other than 'no confidence'. Why is this choice made - does using a higher confidence level strongly impact the results?

R: Since the AOD records with heavy rainfall are not sufficient, to increase the rainfall sample size we chose the data with marginal or higher confidence. We have tested the result using higher confidence, which is similar but not significant as the result in this study.

L159 - Why is a different reanalysis used for the clouds and the meteorology?

R: For meteorology (wind, temperature, humidity etc.), we used both MERRA2 and ERA-interim reanalysis data for consistency. Since ERA-interim reanalysis data does not have three dimensional cloud variables, we used MERRA2 data to examine the cloud effect. However, the latter was taken out of the revised manuscript because they are simulated rather than observed clouds.

L167 – I am not clear why focusing on this time period better identifies the effect of aerosol

R: The reasons for choosing this period are already given in our earlier study (Zhou et al., 2018). Besides the large-scale dynamics, major consideration is that the period has convective rainfall with heavy pollution. To benefit the readers, we have revised the sentences; see Lines 205-208, Page 7.

L174 - These are very high values of AOD. Brennan et al. (2005) suggested that at AOD>0.6, aerosol is likely to be misclassified as cloud. Might this affect the results here?

R: We have been very careful about this issue. First, there are only a few days with the AOD less than 0.6 over BTH region. And we have tried selecting the samples in which AOD<1.0 to do the same analysis using the percentile method and found the result is similar as shown in Fig. A3. Comparing with AOD <0.57 (which is the 25$^{th}$ percentile in AOD < 1.0), the start time and peak time of heavy rainfall is earlier, and the duration is longer when AOD >0.85 (which is the 75$^{th}$ percentile in AOD < 1.0). Besides, the using of CDNC in the revised manuscript can make up for the uncertainties of AOD, thus we did not address this issue in the article.

[Figure]

Figure A3. PDF of (a) start time (units: LST), (b) peak time (units: LST), and (c) duration (units: hours) of heavy rainfall on selected clean (blue lines) and polluted (red lines) conditions when AOD less than 1.0.

L190 - There also appears to be a later peak in the precipitation rate at high AOD. Is it clear whether the peak has move earlier or later.

R: Actually there is also a later peak in the PDF of heavy rainfall start time at high AOD, which is at early morning. While in this study, we mainly focus on the heavy rainfall occurred during afternoon and night, since the heavy rainfall occurred in this time is mostly generated by local convection, while the early-morning rainfall might be associated with the mountain winds (Wolyn et al., 1994; Li et al., 2016) and the nighttime low level jet (Higgins et al., 1997; Liu et al., 2012). Besides, the average start time of heavy rainfall in high AOD is in advance in Tab.1 in the manuscript, and the result using CDNC also shows the heavy rainfall moves earlier. A sentence is added to Lines 245-248, Page 8.

L206 - Given that much of the paper is about the development of precipitation, it might be good to point out that the cloud properties are measured at the same time as the aerosol. This is stated in the methods section, but a small reminder would be useful.

R: Thanks. Lines 304-305, Pages 9-10 are added.

L260 - It is not clear what 'nearly 350hPa' means. 340 or 360hPa?

R: This part is totally re-written; see Lines 310-313, Page 10.

L287 - Twomey not Towmey

R: Corrected.

L290 - via enhanced collision-coalesence - this is not the mechanism stated in Yuan et al. 2008, where the positive AOD-effective radius relationship is related to changes in aerosol properties.

R: Yes, it was an incorrect citation. Since we used CDNC instead of AOD to investigate the aerosol indirect effect in the revised version, the results have been changed; see Section 6.1.

L295 - The Wegner-Bergeron-Findeisen process can act whenever supercooled liquid and ice crystals co-exist. As long as liquid droplets exist, it should not depend directly on the supersaturation over liquid, although if the region is supersaturated with respect to liquid, the liquid droplets can also continue to grow.

R: We are intrigued by the feature that the ice cloud effective radius is decreased when both CDNC and cloud water are increased, and hypothesize that it may be related to the Wegner-Bergeron-Findeisen process. Certainly, modeling studies may provide insights.

L319 - These changes would shift the PDF of CF, but I am not sure it can be said that BC 'corresponds to a slight decrease of CF when CF is more than 90%' as the CF in an aerosol-free atmosphere is not known. Instead, it would be more accurate to use phrases such as 'cloud fractions larger than 90% are less common in high BC environments'.

R: Yes, we have revised the writing; see Lines 415-416, Page 13.

[revised manuscript text omitted]

---

## Author Response (AR2)

**Answers to Referee #1's comments**

**We appreciate very much the constructive comments the reviewer provided, all of which are addressed. The one-to-one correspondence is given below in black.**

*The authors have done a lot of work redoing the paper, and I applaud their effort.*

*I have two critical concerns with this paper:*

*1. Moisture and CDNC/AOD covary in the sample. No multivariate analysis is undertaken and it is possible that covariability between CDNC/AOD and cloud properties or rain statistics are really just moisture variability covarying with precipitation and cloud.*

*2. The calculation of CDNC here does not acknowledge the numerous sources of retrieval error that may covary with precipitation.*

**R: Since the effects of aerosols and moisture on precipitation are not linear, multivariate analysis may not provide useful information. Nevertheless, on the concern about the moisture and CDNC/AOD covary, we add Sect. 3.3 to address the effects of moisture on the heavy rainfall.**

**We agree that the calculation of CDNC needs to be more rigorous and thus we add limiting conditions of CF/COD/CER when calculating CDNC, and add more discussions about the uncertainties of the calculation, see Lines 184-188 in Sect. 2.1.3 and Lines 493-501 in Sect. 5.2. In addition, we also add discussions about the uncertainties of other indicators and associated results in Sect. 5.2.**

*My central concern- and what may be a critical error in the analysis- is that moisture and pollution covary. The authors reveal this in a later part of the paper. To me this potentially means that all their univariate analysis of rain statistics and cloud statistics as a function of CDNC and AOD are spurious and may just be aliasing onto meteorology. This may not be the case once variability in meteorology is better accounted for, but the authors need to state that moisture and aerosol covary in this region right away and then do all of their analysis in a multivariate way (for example, showing variables as 2D PDFs in the space of moisture and CDNC/AOD/AI instead of 1-D PDFs in the space of just CDNC/AOD/AI).*

**R: The analysis of rain statistics is not spurious, as fully discussed in our previous study (Zhou et al., 2018) of the sensitivity of rainfall associated with pollution to the moisture and wind speed. In any case, in the revised manuscript we have: added a section to compare the rainfall behaviors associated with moisture and pollution respectively, and, follow the suggestions, illustrated the scatter distribution of moisture-rainfall characteristics relationship of clean and polluted cases (Fig. 6b&c, equivalent to 2D PDF) and discussed their difference, see Sect. 3.3.**

*The authors have added CDNC, but their approach to this does not incorporate knowledge of the various sources of systematic error in CDNC retrievals from MODIS that we know of (Daniel P. Grosvenor et al., 2018). The authors need to expand their discussion of CDNC, clarify exactly how they are calculating this quantity, and discuss how known covariances between cloud habit, rainfall rate, and CDNC retrieval error might impact their results. This will substantially impact section 5.1 of the MS, but needs to be done as there has been shown to be substantial covariability between cloud morphology and CDNC retrieval error (D. P. Grosvenor & Wood, 2014).*

**R: We agree and have added discussions on some uncertainties, see Lines 493-501 in Sect. 5.2.**

*The incorporation of OMI aerosol index is interesting, and supports their work. I do not know anything about these products, but it is less clear how spurious covariability between these retrievals and rainfall could occur than with MODIS AOD/CDNC. I suggest that they incorporate these measurements into their evaluation of rainfall statistics.*

**R: As shown below, the relationship between aerosol index from OMI and AOD from MODIS show a weak positive correlation. With regard to the co-variability, we have tested the result when removing the cases with AOD more than 2.00, the polluted cases. It shows the consistent result that the heavy rainfall occurs earlier on absorbing aerosol days and the duration is a little longer on scattering aerosol days. We addressed this issue in Sect. 5.2, see Lines 505-508.**

[Figure]

Figure A1. Relationship between AI from OMI and AOD from MODIS on the southwesterly days. The bold line stands for the regression and the thin lines are the thresholds of AI used in the study.

[Figure]

Figure A2. PDF of (a) start time (units: LST), (b) peak time (units: LST), and (c) duration (units: hours) of heavy rainfall on the days that SAI more than 75$^{th}$ tercile (blue lines) and days that AAI more than 75$^{th}$ tercile (red lines) when AOD is less than 2.00, during early summers from 2005 to 2012.

*Overall, I think the paper has potential in the use of ground-based data to get around shortcomings in remote-sensing of rain, but some work has to be done on the analysis to acknowledge the information we have about CDNC retrievals and to work up from the assumption that meteorological variability is the leading order driver of cloud and precipitation variability- as opposed to putting it in as an afterthought.*

*The paper has some difficulty with grammar, but generally the intent of the authors is understandable and I am assuming writing will be tuned up by the ACP proofing staff.*

**R: We have tried our best to edit the manuscript.**

*Ln 36: Do the authors mean this paper, or a recent paper they have published?*

**R: We mean a recent paper we have published (Zhou et al., 2018). The sentence is revised, see Line 36.**

*Ln 47: Correlation and causation are not the same thing. Changes in cloud statistics covary with the proxies for aerosol used in this study.*

*Ln 51: It is unclear what this sentence means.*

*Ln 47: It is unclear what these numbers mean. What was the change in aerosols (AOD/CDNC?)*

**R: We have re-written the abstract, see Lines 46-51.**

*Ln 84: What is moisture supply? Is this the RH at some pressure level? I see this isn't described until line 188.*

**R: We use mean specific humidity at 850hPa (absolute humidity at 850hPa in the last version). This is explicitly mentioned in the abstract and introduction, see Lines 49-50, 105-106.**

*Ln 134: I would say that it tends to be before heavy rainfall events, however, I find Fig. 2 a little confusing- where did LST 6-13 go? Where there never any rainfall events in that time range? That seems unlikely. If I linearly interpolate between the last point and the first point I can see that there a fair number of rainfall events occurring at 10:30 LST, although not the maximum of the PDF- so I would say 'tends' and not 'always'.*

**R: To better match other satellite data and meteorological data, we count "a day" to be from 8 LST to 8 LST next day (0 UTC – 24 UTC). The Terra satellite overpass the BTH region is at around 10:30 LST and when it overpasses BTH, the data is almost missing when it is rainy at that time. So it is not that there are never any rainfall events, rather the rainfall data at that time is removed when we use satellite data. In addition to the missing data, it shows that there are less rainfall cases starting in the morning. Shown in the figure below, the only difference between the two lines is that the latter added ' if (.not. is missing(AOD))' in the code. We revised the statement, see Lines 139-142 in Sect. 2.1.2**.

[Figure]

Figure A3. Diurnal variations of rainfall amount (0.1mm/hour) on the total heavy rainfall days (blue line) and heavy rainfall days with AOD record (red line).

*Ln 167: What band is used to retrieve reff? It makes a big difference (Daniel P. Grosvenor et al., 2018). Are low (80%) cloud fractions removed? From looking ahead I infer that they are not. This is also important to deal with heterogeneity effects, which are critical since the authors are looking at precipitation (see restrictions applied in (Bennartz, Fan, Rausch, Leung, & Heidinger, 2011)).*

**R: We have added limiting conditions of CF when calculating CDNC as you suggested, and the results are consistent when removing low (less than 80%) cloud fraction. In addition, we have added discussion about the uncertainty of CDNC, see Lines 184-188 in Sect. 2.1.3 and Lines 493-501 in Sect. 5.2.**

*Ln 217: As noted in the previous revision, it is unclear if CDNC can be used to stratify into clean and polluted cloud to learn anything about rain. This is because cloud top heterogeneity affects the retrieval of CDNC from MODIS (see Fig 16 of D. P. Grosvenor and Wood (2014)). For example, higher cloud top heterogeneity could go in step with higher rain rates and higher CDNC. The authors need to discuss this source of spurious covariability.*

**R: We have added limiting conditions of CF/COD/CER when calculating CDNC to reduce the influence of heterogeneity effect, and added discussions about the uncertainties of CDNC in the revised version, see Lines 184-188 in Sect. 2.1.3 and Lines 493-501 in Sect. 5.2.**

*Ln 238: I think this section is compelling, especially combined with table 2, which shows that high AOD and CDNC tend to lead to the same change in sign for changes in precipitation characteristics. I do recommend that the authors look into the difference in log intensity since the distribution of intensity is clearly non-normal. Can a similar table be made for the AI from OMI and SO4/BC AOD? That would be nice to compare across.*

**R: We have revised the intensity in Fig. 2 into log intensity but the changes are also non-significant. A table (Tab. 3) made for AI and sulfate/BC AOD is added in the revised manuscript.**

*Ln 302:*

*I think that this section looking at cloud changes sorted by CDNC could be removed. The retrievals of CDNC, LWP, CF, CER and COT are all dependent on each other and retrieval errors in one will affect all the others. I am not sure how much this analysis adds to the primary results of the paper, which is (in my reading) the covariability between*

*precipitation statistics and CDNC/AOD. However, I understand that this analysis helps the authors support their hypothesized mechanism, so I think that if the retrievals in this section are heavily caveated in that none of the variables are truly independent retrievals then that will be acceptable.*

**R: This section is revised as follows. Both AOD and CDNC are used to investigate the concurrent cloud changes associated with pollution during heavy rainfall days. Only changes of variables such as CTP, liquid & ice CWP, and ice CER that are independent of CDNC retrieval are compared with the results based on AOD. We have revised this part and added discussions about the uncertainties of cloud change based on AOD and CDNC, see Lines 360-368 in Sect. 4.1 and Lines 521-537 in Sect. 5.2.**

*Ln 377: In the last part of the section the authors discuss how moisture and CDNC covary in their sample. This means that most of the paper up till now may be looking at spurious covariability. We would expect that meteorology (in this case- moisture) to dominate driving cloud property variability. This also means that the results relating to rainfall are potentially just looking at times when it is more or less moist.*

*To clarify- if I believe that aerosol has no effect on rain rates, based on what the authors have just told me it seems perfectly reasonable to suspect that meteorology drives everything and that CCN-proxies are just covarying with moisture. Obviously this is an empirical study and they can't fully rule out the possibility that CCN proxies are covarying with some unknown predictor, but the authors need to start out binning by CCN proxy and their meteorological parameter at the beginning instead of introducing it now.*

**R: In our previous study (Zhou et al., 2018), we have indicated that advance of heavy rainfall still exists when limiting the condition of humidity. Classifying the heavy rainfall by AOD/CDNC and moisture at the beginning will lead to the large difference of case numbers between clean and polluted conditions, which causes more uncertainty to rainfall statistics. Hence, to address the issue you raised, we added a section to compare the moisture effect and aerosol effect on the heavy rainfall, see Sect. 3.3.**

*Ln 416: Or estimated inversion strength and BC covary.*

**R: We agree and added this phrase, see Lines 442.**

*Ln 431: Or estimated inversion strength and SO4 covary- or SO4 is bright and leads to enhanced cloud fraction through misdetection.*

**R: We modified the statement, see Lines 458-461 in Sect. 5.1. The misdetection of MODIS is possible, although the MODIS C6 product has progress in detect aerosols and clouds (Levy et al., 2013). We added discussions in Sect. 5.2, see Lines 483-486.**

*Fig 8. Something looks off in this plot and Fig. 6 (top left). The CF doesn't seem to have a very smooth distribution. Why are there so many completely overcast 1x1° days with no transition?*

**R: Since the daily clouds we investigate are all on the heavy rainfall days, we suppose most cases with more than 90% cloud fraction might be reasonable. The misdetection of CF is also possible, and we added discussion about it, see Lines 484-485 in Sect. 5.2.**

*Bennartz, R., Fan, J., Rausch, J., Leung, L. R., & Heidinger, A. K. (2011). Pollution from China increases cloud droplet number, suppresses rain over the East China Sea. Geophysical Research Letters, 38(9), n/a-n/a. doi:10.1029/2011gl047235*

*Christensen, M. W., Neubauer, D., Poulsen, C., Thomas, G., McGarragh, G., Povey, A. C., . . . Grainger, R. G. (2017). Unveiling aerosol-cloud interactions Part 1: Cloud contamination in satellite products enhances the aerosol indirect forcing estimate. Atmos. Chem. Phys. Discuss., 2017, 1-21. doi:10.5194/acp-2017-450*

*Grosvenor, D. P., Sourdeval, O., Zuidema, P., Ackerman, A., Alexandrov, M. D., Bennartz, R., . . . Quaas, J. (2018). Remote Sensing of Droplet Number Concentration in Warm Clouds: A Review of the Current State of Knowledge and Perspectives. Reviews of Geophysics, 56(2), 409-453. doi:10.1029/2017rg000593*

*Grosvenor, D. P., & Wood, R. (2014). The effect of solar zenith angle on MODIS cloud optical and microphysical retrievals within marine liquid water clouds. Atmos. Chem. Phys., 14(14), 7291-7321. doi:10.5194/acp-14-7291-2014*

*Twohy, C. H., Coakley, J. A., & Tahnk, W. R. (2009). Effect of changes in relative humidity on aerosol scattering near clouds. Journal of Geophysical Research: Atmospheres, 114(D5), n/a-n/a. doi:10.1029/2008JD010991*

Levy, R. C., Mattoo, S., Munchak, L. A., Remer, L. A., Sayer, A. M., Patadia, F., and Hsu, N. C.: The Collection 6 MODIS aerosol products over land and ocean, Atmos. Meas. Tech., 6, 2989–3034, https://doi.org/10.5194/amt-6-2989-2013, 2013.

Zhou, S., Yang, J., Wang, W. C., Gong, D., Shi, P., and Gao, M.: Shift of daily rainfall peaks over the Beijing–Tianjin– Hebei region: An indication of pollutant effects? Int. J. Climatol. 2018:1–10. https://doi.org/10.1002/joc.5700, 2018.

**Answers to Referee #2's comments**

**Thanks for the helpful comments. The point-by-point response is given below in black.**

*I thank the authors for the work they have put into revising the paper, it is much improved over the previous version, although there are a number of things that I am still concerned about.*

*The linking of aerosol properties to the diurnal variation of aerosol remains a nice result to base this study around, but the correlation between cloud and aerosol properties shown in later parts of the work is subject to a number of meteorological covariations. Just controlling for the humidity has been shown to be insufficient to isolate an aerosol effect (Quaas and Boucher, 2012). With this in mind, Section 5 is useful in enumerating the cloud properties that are correlated to CDNC, but I am not sure it demonstrates a causal role of aerosol. I am not sure that it is vital to explain the rainfall results from the first part of the paper. Perhaps this section could be presented relationships between cloud and aerosol properties, rather than evidence of an aerosol impact.*

**R: We agree, and the subtitles and some statements in Sect. 5 (now Sect. 4 in the revised manuscript) have been revised.**

*Main points:*
*I like the use of the CDNC as an aerosol proxy, but I think that the authors need to do more to demonstrate that it is applicable in this case. It is not just a drop-in replacement for AOD and has a number of specific biases.*

*The CDNC calculation is only applicable in adiabatic clouds. In general, convective clouds are not adiabatic and a precipitating cloud is unlikely to be. This may not be a large factor in this study, but it should be considered and factored into the discussion/conclusions. There are a number of other potential biases that may affect this work. Grosvenor et al. (2018) is a good summary.*

*Additionally, is the CDNC calculation done using the 1 by 1 degree mean data? As a non-linear calculation, this can strongly impact the ``retrieved'' CDNC. The MODIS L3 product includes a cloud optical depth-effective radius joint histogram which can be used to better calculate the CDNC (e.g. Quaas et al., 2008; Grandey and Stier, 2010)*

**R: The references are useful. We agree that large uncertainties exist for the derived CDNC (Daniel P. Grosvenor et al., 2018). The CDNC in the study is calculated using 1° x 1° data from MODIS L3 product. As indicated by Grosvenor et al. (2018), the uncertainties in CDNC from MODIS with 1° x 1° spatial resolution can be 54% or even more. However, we assume the relative differences of CDNC at different time and location should be with much less uncertainties, and they are the main information we adopted. To make the calculation of CDNC more rigorous, we have added limiting conditions of CF/COD/CER when calculating CDNC to reduce the uncertainties, and we added some discussions about the uncertainties of the calculation, see Lines 185-188 in Sect. 2.1.3 and Lines 493-501 in Sect. 5.2.**

*As the CDNC is calculated from the CER and COD, it is not clear to me that the CDNC-CER and CDNC-COD relationships are useful. I have similar concerns about the CDNC-LWP relationship, which is difficult to interpret even in low-level liquid clouds (e.g. Sato et al, 2018; Gryspeerdt et al., 2018b). In this case, I think that Fig. 7 is just reproducing the assumptions used to calculate the CDNC and LWP (if the CER and CDNC are known, then the LWP is uniquely identified - at least at a pixel level).*

**R: The cloud section is revised to use both AOD and CDNC indicators to investigate the concurrent cloud**

**changes during heavy rainfall days. Only changes of variables such as CTP, liquid & ice CWP, and ice CER that are independent of CDNC retrieval are compared with the results based on AOD. We have revised this part and added discussions about the uncertainties of cloud change based on AOD and CDNC, see Lines 360-362 in Sect. 4.1 and Lines 532-537 in Sect. 5.2. And Fig.7 is deleted.**

*Finally, is it clear what biases in the CDNC might be caused by the addition of thin overlying ice cloud? The authors are considering very complex situations where this may be an issue in a way that it is not for studies of liquid clouds.*

**R: Since the liquid CER and liquid COD from MODIS which calculate the CDNC might be affected by overlying ice cloud, the CDNC might be affected as well. We added discussion about it in the revised version, see Lines 483-486 in Sect. 5.2.**

*Minor points:*
*L188 - What are the advantages of the AH here compared to a more common meteorological value such as specific humidity (easily available from ERA-Interim)?*
**R: We have changed the AH into SH in the revised manuscript, their results show consistent features.**

*L218 - Absorbing aerosol index is dependent on the altitude of the aerosol layer. What is assumed in this work?*
**R: We have compared the difference of aerosol profiles obtained from MACC reanalysis data between selected absorbing aerosol days and scattering aerosol days, although the profiles from MACC may be not authentic. The profiles of different aerosols (BC, OC, sulfate and dust) show similar characteristics that the maximums occur at the similar altitude (around 900-1000 hPa). Hence, we assume that the altitude of aerosol layer has almost no influence on the results analyzed by aerosol index in the study.**

*L406 - How does decreasing the supersaturation reduce the strength of the freezing process? The supersaturation over ice is higher than over liquid. There are some studies which have noted an aerosol relationship to observations of mixed phase and ice cloud that might help explain this result (Chylek et al., 2006; Zhao et al., 2018; Gryspeerdt et al., 2018)*
**R: Thanks for the references. We have revised the explaination, see Lines 526-528 in Sect. 5.2.**

*L446 - There have been many studies looking at the impact of BC on precipitation, perhaps they might be helpful in interpreting these results (e.g. Fan et al., 2008; O'Gorman et al., 2011). The role of BC is expected to change with altitude. Is it clear that the BC here is all in the boundary layer?*
**R: Thanks for the references. Since reliable observational aerosol vertical data are not available, we use MACC and MERRA2 reanalysis data (Fig. 3e in Zhou et al., 2018) to provide this information: the height of BC is around 700-1000 hPa (around 0-3000m from surface). The boundary layer height on the polluted days (based on AOD) is below 1600 m in average during daytime thus there might be BC above the boundary layer. Our results show that under pollution, the upward motion is enhanced, the boundary layer is more stable but the lower to middle atmosphere (850-500 hPa) is more unstable, which may be related to the heating effect of BC (Zhou et al., 2018).**

*L462 - This might be due to a change in LWP/cloud depth with changing AOD. However, it is not clear if that change in cloud depth could be attributed to aerosols.*
**R: We agree and revised the statements, see Lines 534-537 in Sect. 5.2.**

*Figures - The text on many of the plots is very small.*
**R: We have modified the plots.**

[revised manuscript text omitted]

Siyuan Zhou 19/5/15 9:46 AM
已删除: (Fig. 6). Therefore, the rainfall start time is delayed for

Siyuan Zhou 19/5/15 9:46 AM
已删除: reduced collision-coalescence of cloud droplets, while the

Siyuan Zhou 19/5/15 9:46 AM
已删除: might be prolonged due to the significant increase of cloud water

Siyuan Zhou 19/5/15 9:46 AM
已删除: associated with the scattering aerosols (sulfate)

Siyuan Zhou 19/5/15 9:46 AM
已删除: mainly

Siyuan Zhou 19/5/15 9:46 AM
已删除: .

Siyuan Zhou 19/5/15 9:46 AM
已删除: using

Siyuan Zhou 19/5/15 9:46 AM
已删除: and

Siyuan Zhou 19/5/15 9:46 AM
已删除: .

Siyuan Zhou 19/5/15 9:46 AM
已删除: radiative effect and cloud

Siyuan Zhou 19/5/15 9:46 AM
已删除: of aerosols

**5.2 Uncertainties of different indicators and associated distinct results**

The gauge-based hourly precipitation data used in this study is more reliable than other observational and reanalysis precipitation data. In contrast with precipitation datasets, the observation of aerosols and clouds from MODIS might have larger uncertainties, e.g., which come from the misdetection of CF when AOD is large (Brennan et al., 2005; Levy et al., 2013) or the mutual interference between liquid and ice clouds (Holz et al., 2008; Platnick et al., 2017).

In this study, we used two pollution indicators, AOD and CDNC, which discriminates the pollution levels for different purposes. AOD is a good proxy for the large-scale pollution level, but it stands for the optical feature of aerosols and cannot well represent CCN when we studied the aerosol-cloud interaction (Shinozuka et al., 2015). The value of AOD is also influenced by moisture condition, which is aerosol humidification effect (Twohy et al., 2009; Altaratz et al., 2013). Therefore, we comprehensively analyzed the moisture effect on the results. CDNC is a better proxy for CCN, which facilitates the study on the cloud changes associated with aerosol pollution. But the retrieved CDNC has larger uncertainties. First, the assumptions in the calculation of CDNC are idealized that CDNC is constant with height in a cloud and cloud liquid water increases monotonically at an adiabatic environment (Grosvenor et al., 2018). Second, as indicated by Grosvenor et al. (2018), the uncertainties in the pixel-level retrievals of CDNC from MODIS with 1°x1° spatial resolution can be above 54%, which come from the uncertainties of parameters and the original COD and CER data using in the calculation, and also the influence of heterogeneity effect from thin clouds. To reduce the influence of heterogeneity effect as much as possible, we have attempted to limit the conditions of CF, liquid COD and CER when calculating CDNC in the study. Besides, this study primarily applied the relative changes of CDNC, which actually reduced the uncertainties of absolute values.

We applied ultraviolet AI and AOD of BC/sulfate to identify different types of aerosols. The AI datasets from OMI, which can distinguish the absorbing aerosols and scattering aerosols, also have uncertainties especially for the near-zero values. Hence, we only compare the extreme circumstances of absorbing aerosols and scattering aerosols. We also found the AI has a weak positive correlation with AOD from MODIS, which indicates the results on absorbing aerosol days might represent the results on polluted days if identified by AOD. To avoid the uncertainty, we re-examined the results using AI when removing the polluted cases identified by AOD, and found the major results are not changed. The comparisons of BC/sulfate AOD cases also have uncertainties because they are retrieved from MACC reanalysis data. Although the above four indicators have their own uncertainties, currently we cannot find more reliable datasets in a long-term observational record, and the major findings can be well shown in these four indices.

Using AOD and CDNC we have drawn the same conclusion that the heavy rainfall occurs in advance and the duration is prolonged under pollution (Fig. 2). We found the AOD and CDNC only have a weak positive

Siyuan Zhou 19/5/15 9:46 AM
已删除: 6.3
Siyuan Zhou 19/5/15 9:46 AM
已删除: ,
Siyuan Zhou 19/5/15 9:46 AM
已删除: to discriminate the different
Siyuan Zhou 19/5/15 9:46 AM
已删除: , which are AOD and CDNC.
Siyuan Zhou 19/5/15 9:46 AM
已设置格式: 字体颜色: 自动
Siyuan Zhou 19/5/15 9:46 AM
已删除: ).
Siyuan Zhou 19/5/15 9:46 AM
已删除: but it also
Siyuan Zhou 19/5/15 9:46 AM
已删除: its uncertainties because it is calculated
Siyuan Zhou 19/5/15 9:46 AM
已设置格式: 字体颜色: 文字 1
Siyuan Zhou 19/5/15 9:46 AM
已删除: the COT
Siyuan Zhou 19/5/15 9:46 AM
已设置格式: 字体颜色: 文字 1
Siyuan Zhou 19/5/15 9:46 AM
已删除: . We can draw the same conclusion on heavy rainfall diurnal changes between clean and polluted condition when
Siyuan Zhou 19/5/15 9:46 AM
已设置格式: 字体颜色: 文字 1
Siyuan Zhou 19/5/15 9:46 AM
已删除: AOD and CDNC respectively (Fig. 2). But when investigating the differ ... [40]
Siyuan Zhou 19/5/15 9:46 AM
已删除: is decreased
Siyuan Zhou 19/5/15 9:46 AM
已设置格式: 字体颜色: 文字 1
Siyuan Zhou 19/5/15 9:46 AM
已删除: CDNC increases (Fig. 6) w ... [41]
Siyuan Zhou 19/5/15 9:46 AM
已设置格式: 字体颜色: 文字 1
Siyuan Zhou 19/5/15 9:46 AM
已设置格式: 字体颜色: 文字 1
Siyuan Zhou 19/5/15 9:46 AM
已删除: Ultraviolet
Siyuan Zhou 19/5/15 9:46 AM
已删除: in this study is
Siyuan Zhou 19/5/15 9:46 AM
已删除: used to detect
Siyuan Zhou 19/5/15 9:46 AM
已删除: since the near-zero values ... [42]
Siyuan Zhou 19/5/15 9:46 AM
已删除: all
Siyuan Zhou 19/5/15 9:46 AM
已删除: the

[revised manuscript text omitted]

Siyuan Zhou 19/5/15 9:46 AM
已删除: . ... [43]

Siyuan Zhou 19/5/15 9:46 AM
已删除:

| Indicator | Da |
|---|---|
| AOD | M |
| CDNC ($cm^{-3}$) | M |
| AAI | |
| SAI | |
| BC AOD | M |
| Sulfate AOD | M |
| Absolute humidity at 850 hPa ($g/m^3$) | ERA |

Siyuan Zhou 19/5/15 9:46 AM
已删除:

| Characteristics of heavy rainfall | Average of c condition | |
|---|---|---|
| | AOD | C |
| Start time (LST) | 24.2 | 2 |
| Peak time (LST) | 23.0 | 2 |
| Duration (hours) | 4.0 | |
| Intensity (0.1mm/hour ) | 164.9 | 1 |

Siyuan Zhou 19/5/15 9:46 AM
已删除: condition

Siyuan Zhou 19/5/15 9:46 AM
已删除: condition

Siyuan Zhou 19/5/15 9:46 AM
已删除: of differences

[revised manuscript text omitted]

---

## Author Response (AR3)

**Responses to Referee #1's comments**

*The authors have addressed my two major concerns from the last review and have done a considerable amount of work showing robustness in their results. Overall I find the paper acceptable for publication and I commend the authors on all their hard work to untangle aerosol-precipitation effects in this interesting study.*

*The segregation of data into polluted/clean moist/dry days is very compelling and I appreciate their additional work on the arduous work of calculating CDNC.*
**Reply**: Thanks.

*Figure 6 is very compelling. The authors could consider using a 2d-bivariate kernel density estimate (https://seaborn.pydata.org/generated/seaborn.kdeplot.html) to visualize the density of the relatively sparse data points for additional punch to the figure, but this is totally optional and I find the figure convincing as is.*
**Reply**: Thanks for suggesting this method. However, the 2d-PDF plot overlaps and not as clear as Figure 6, so we keep Figure 6 as it is.

*One thing that would be interesting (but not essential) would be to examine if total rainfall over a day if affected by CDNC, or if it's just onset, etc. That is to say, does accumulated rainfall care at all about CDNC, or is it just driven by moisture? If this is not the case this would be worth noting for comparison to other studies.*
**Reply**: Intuitively, both the CDNC and moisture affect the total rainfall throughout the day. Table below shows the total rainfall over a day categorized individually by CDNC and specific humidity at 850 hPa. It therefore implies that moisture has larger influence on the daily rainfall amount.

| Total rainfall over a day (mm) | CDNC<25th | CDNC>75th | SH<25th | SH>75th |
|---|---|---|---|---|
| Average for heavy rainfall | 29.3 | 33.9 | 25.6 | 35.6 |
| Average for total rainfall | 11.8 | 12.4 | 7.2 | 17.0 |

Table S1. Averaged total rainfall over a day (mm/day) classified by CDNC and SH.

*Table 1-5. Might be good to give standard error on the means. Not essential.*
**Reply**: We have added standard deviations in Table 2-5.

**Responses to Referee #2's comments**

*The authors have done a significant amount of work improving the manuscript. The precipitation part of this paper is interesting and I think worthy of publication. However, some of my concerns remain, particularly in regards to the detection of causal relationships and the use of the CDNC.*

**Aerosol-cloud-precipitaiton relationships**

*I understand that it is difficult to control for meteorological covariations and systematic biases when performing these observational studies and I appreciate the extra section on the impact of humidity that the authors have included. Section 3.3 (and 4.2) is a useful addition to the paper, especially given that the paper is dealing with a difficult subject. However, it seems that the results from section 3.3 do not make it to the abstract and conclusions, which still give the impression that aerosols have been shown to cause these changes in cloud and precipitation properties. Where these results from section 3.3 are mentioned (e.g. in the abstract, L49-51), it is almost as an afterthought, failing to point out that the impact of moisture changes is the same as the impact of sulphate, which means that the aerosol effect cannot be isolated from the impact of humidity.*

**Reply**: Yes, it is our oversight, and now the abstract and conclusions cover these points reminded by the Reviewer, see L45, 49-50, 559-562.

*It has previously been shown that using reanalysis moisture variables cannot completely control for meteorological covariations between aerosol and precipitation (Boucher and Quaas, 2012). In addition, several of the relationships examined within section 4 are known to be affected by meteorological covariations: AOD-CF (Quaas et al., 2010; Grandey et al., 2013) and AOD-CTP (Gryspeerdt et al., 2014). Where these relationships are discussed in this paper (e.g. L521 onwards), they appear to be used as evidence for an aerosol effect on cloud, despite these known issues.*

**Reply**: We agree, and have re-written the text to reflect the points, see L540-546 in section 5.2.

*This is not a large change to the paper, mostly just in the abstract and conclusions (e.g. changing sentences such as L547 - '... the different roles of [aerosol] in modifying the diurnal ...' to '... the different relationships of [aerosol] to the diurnal ...'). I also would suggest that the discussion in section 5, particularly around the relationships between AOD and cloud properties, is modified to reflect the previous results.*

**Reply**: See above.

**CDNC retrieval**

*My fourth point in the previous review mentioned that it is not clear that the CDNC-CER and CDNC-COT relationships are useful, as they are not independent. The response suggests that these have been removed, but investigation of the CDNC-CER relationship is still present and compared with the AOD-CER relationship (e.g. L360, L521, L532). There may be a justification for including the CDNC-CER relationship (I am not clear that it has to be removed), but could the authors clarify whether they are removing it or not?*

**Reply**: We agree and have removed the CDNC-CER/COT versus AOD comparison in sections 4.1 and 5.2. Because we believe that the increased liquid COT and decreased liquid CER may be not completely caused by CDNC calculation but the effect of CCN (which is explained in section 4.2 L407-409), these results remain in the figure 7 and table 4 for reference.

*It is good to include the reference to the Grosvenor paper and a discussion of the uncertainties. However, I am not sure the impact of the uncertainties on the results is addressed. In particular, the CDNC retrieval is for adiabatic clouds, and it is not clear that it can be applied to convective clouds. Clouds that are raining are by definition non-adiabatic. How does this affect the results in this work? Might it reduce the significance of the results?*

**Reply**: The use of CDNC is to provide corroborative evidence for the results based on AOD. We understand the issue and uncertainties of "adiabatic" clouds, however, we believe that it does not affect the major findings of this study. Anyway, we added additional statement to highlight it, see L496-497.

*Section 4 starts referring to the CDNC as CCN. Is this appropriate? CDNC depends on CCN and updraught, presumably there is a strong variation in updraught between convective clouds which weakens the link between CDNC and CCN?*

**Reply**: In this study, we use CDNC as a surrogate for CCN, while the issue of updraught, although important, is outside the scope of this study. Nevertheless, for clarification, we have modified the statements in section 4.

*The average values for the retrieved CDNC (L391) are very high (more than 2000 cm-3). For an optical depth of 100, this would mean that all of the effective radius retrievals are smaller than around 6um. Is this correct?*

**Reply**: This question prompted us to double check the results. As it turns out that we made an error in the CDNC unit conversion from the units of $C_w$, $\rho_w$, and $R_e$ having $gm^{-4}$, $kgm^{-3}$, and μm respectively. So the value of CDNC (=2000) should be reduced by $10^{3/2}$ ($10^{15/2}$ versus $10^9$) cm$^{-3}$ to a value of 63.2 cm$^{-3}$, and we have revised the values in L390-391 and Table 1. This error nevertheless does not affect other results. In any case, we deeply appreciate this comment.

*Following my third point in the previous review, it is not clear how the CDNC is calculated. Are the 1 by 1 degree mean values of CER and COT used? If so, this should be noted, as it may cause an underestimate in the CDNC compared to other studies (e.g Quaas et al., 2008;*

*Grandey et al., 2010) which use the level 2 data or the L3 joint histogram 'Cloud_Optical_Thickness_Liquid_JHisto_vs_Eff_Radius' to calculate the CDNC.*

**Reply**: We used 1 by 1 degree mean values of CER and COT of L3 cloud product of MODIS. We have clarified the data use in L182-184. However, we are not sure if the CDNC is underestimated since both COT and CER may be underestimated.

*Minor points*

*L442 - Ackerman (2000) is a usual reference for the semi-direct effect.*
**Reply**: Thanks for the reference, we have added it (L443).

*L470 - 'absorbing aerosol effect' -> 'absorbing aerosol results'? I am not sure that an effect is specified (to me, it would require some kind of process being suggested)*
**Reply**: We have modified the statements, see L300, 302-303, 470-471.

*Section 4.2 - repeated references to section 5.1, when presumably 4.1 is meant?*
**Reply**: I understand what you mean is in L398 and L433 that 5.1 should be 4.1, we have revised it.

*L500 - Relative changes are still subject to systematic biases, which are the more important source of error in a study like this.*
**Reply**: We have modified the sentence, see L502-504.

*L510 - I understand that these datasests are the best available. However, if they are insufficient to demonstrate a causal effect of aerosol on cloud or precipitation properties, then they are insufficient. In that case, this must be noted.*
**Reply**: We agree and added sentence about it, see L512-516.

*L520 - Or sampling differences between the AOD and CDNC?*
**Reply**: We agree and added it, see L524.

*L539 - What happened to section 6?*
**Reply**: Revised, see L548.

*Fig. 8 - The units for the x-axis are "%" only*
**Reply**: Revised, see Figure 8.

*There are still some spelling/grammar issues (e.g. black carbons - L44 and in the new sections), but they don't appear to make to too difficult to understand the paper and could be corrected in proof reading.*
**Reply**: We have carefully checked the manuscript again and corrected some mistakes.

[revised manuscript text omitted]

思媛 周 19/7/21 12:02 AM
已删除: aerosols impact

---

## Author Response (AR4)

**Responds**

**To accommodate the comments from Reviewers and Editor, we have again made major revisions to the manuscript, as first summarized here:**

**1. We adjusted the structure of the article especially the "discussion" section to make it clearer and logical. The paper is re-organized as: The data and methodology are introduced in Sect. 2. Section 3 addresses the relationship between aerosol pollution and diurnal variation of heavy rainfall, covering the distinct characteristics of heavy rainfall using AOD and CDNC, the different behaviors of heavy rainfall along with different types of aerosols, and the comparison of heavy rainfall behaviors influenced respectively by moisture and aerosols. Section 4 describes the concurrent changes of cloud properties associated with aerosols and compares the possible influences of CDNC (CCN) and moisture on the cloud properties. Section 5 gives the hypothesis about the mechanisms of aerosol effects on the heavy rainfall. Conclusions and discussion are given in Sect. 6.**

**2. We have added additional results about the influence of moisture, such as the comparison in the influences of sulfate aerosols and moisture on the heavy rainfall and the results of rainfall characteristics when removing the high moisture samples in Sect. 3.3, as well as the results and discussions of cloud properties on different moisture conditions in Sect. 4.1&4.2. We also modified the contents concerning moisture in the abstract, conclusion and discussion.**

**3. We have added descriptions about the distribution of AOD and CDNC in Sect. 3, and other discussion that Reviewers raised, including the limitation of CDNC as surrogate of CCN concerning the updrafts and the uncertainties from the comparison between the extreme circumstances in Sect. 6.2.**

**More details are presented below:**

*Response P2:*
*A major issue of the manuscript remains that it is predicated on the assumption that the derived aerosol-cloud relationships are causal. Some caveats have been inserted in response to critical reviewer comments, e.g. related to relative humidity, but the overall wording of the manuscript remains fairly one-track and I agree with the reviewer that this appears to be more an afterthought. In a revised version of the manuscript, it is critical to separate the description of the derived relationships (e.g. for higher AOD the onset of precipitation occurs…) from a causal attribution (e.g. aerosols delay the onset of precipitation). Causal attribution needs to be presented and discussed in the context of confounding variables. Potential limitations of all causal attributions should be explicitly discussed. This critical analysis needs to be pulled through to the abstract and conclusions.*

**R: We have modified the whole structure and contents including the abstract and conclusions to make**

it more comprehensive and objective as you mentioned, and added more results and discussions on the moisture influence in Sect. 3.3, 4.1 and 4.2. For example, we have tested the sensitivities of the changes of heavy rainfall and clouds associated aerosols to moisture in Sect.3.3&4.1, and compared and discussed the influences of sulfate/CCN and moisture on the rainfall and clouds in Sect. 3.3&4.2.

*Response P3:*

*The use of CDNC as surrogate for CCN has severe limitations and it is not sufficient to simply declare the issue of updrafts out of scope of this study. Aerosol activation is controlled by CCN and updraft velocity. The region of interest is highly polluted so this is very likely to be an updraft limited regime (c.f. Reutter et al., ACP, 2009). Assuming this is the case, any presented correlation with CDNC with cloud properties could be interpreted as correlation of updraft velocity with the respective cloud property or precipitation – which are of course be expected, even under constant CCN. There are good reasons to believe that this could significantly contribute to or even dominate your presented correlations. If this analysis is retained, its reliability needs to be demonstrated. This cannot be ignored in the analysis and discussion.*

**R: We have added the discussion about the limitations of the use of CDNC as surrogate for CCN concerning the influence of the updraft (vertical velocity). We suppose that the CDNC could still represent the CCN for that: first, the results show that there is no significant correlation between CDNC and vertical velocity although the latter is larger in the polluted cases based on CDNC, and the percentage of the cases with CDNC more than 75[th] percentile and updraft more than 75[th] percentile in the cases only with updraft more than 75[th] percentile is only 30.6%. Second, the results of heavy rainfall characteristics based on CDNC do not change when limiting the vertical velocity to a certain range (less than 25[th] percentile, 25[th] – 75[th] percentile or more than 75[th] percentile). Third, we think the increase of vertical velocity on the polluted days is related to the increase of pollutants (Zhou et al., 2018), which means the pollutants including the aerosols serving as CCN and updraft are co-varied and they might be the cause and effect for each other. Therefore, we suppose the CDNC could stand for CCN to a certain extent although it might be partially dependent of updraft velocity, see Lines 592-604 Page 18.**

*Response P3:*

*You have now corrected your retrieved CDNC values by more than an order of magnitude from 2000cm-3 to a new value of 63cm-3, which seems unrealistically low for such a polluted region (c.f. Grosvenor et al., 2018). It appears neither of these values have been evaluated or critically analysed. This needs to be done thoroughly for this data to be used in the manuscript.*

*Given the significant structural uncertainties involved it is not appropriate to conclude on upper or lower bounds of the total effect from a single modelling study. Even if structural errors were negligible there exist a large number of parametric uncertainties that have not been sampled. The only dimension you sample is the perturbation strength so this need so be clear. However, given the extreme range of values chosen, these*

*perturbations are likely to act as on/off switch for some processes, such as autoconversion. The implications should be discussed.*

**R: We have carefully double checked the calculation of CDNC and compared our values with other studies. The averaged CDNCs on the non-rainfall days, rainfall days and heavy rainfall days during 2002-2012 early summers are 54.70, 72.92, and 68.66 cm$^{-3}$ respectively according to our calculation, which are similar with the global mean value 71.0 ± 26.4 cm$^{-3}$ based on MODIS C6 product using 3 months spanning JJA, 2008 (Grosvenor et al., 2018). Although our values seem a little smaller for this polluted region, we think the distribution and the range (around 20-500 cm$^{-3}$) of the CDNC is reasonable when we look at the spectral distribution of CDNC (Fig. 2 in the manuscript). We admit there might be systematic biases in the calculation of CDNC in our study, but we think the values are acceptable after the check of the calculation and the comparisons with other studies (e.g. Zhu et al., 2018). We have added the description about the distributions and ranges of AOD and CDNC in Sect. 3, see Lines 251-264 Page 8.**

**It makes sense that comparing the extreme conditions of AOD and CDNC could bring uncertainties that these extreme conditions might be related with totally different microphysical process or meteorological background. Hence, to reduce the uncertainty, we further checked the results of heavy rainfall using the middle range of AOD and CDNC such as 25$^{th}$ – 50$^{th}$ percentile versus 50$^{th}$ -75$^{th}$ percentile. The results are basically the same and significant except the change of the peak time of heavy rainfall is not significant based on AOD. We added discussion about it, see Lines 625-632 Page 19.**

*Figures & Tables*
*All figures and tables need to be fully self-explaining and captions need to state the used data sources.*

**R: Thanks, modified and added.**

**References:**

[revised manuscript text omitted]

Siyuan Zhou 19/10/9 9:29 PM
已上移 [6]: the increased cloud dr... [16]

Siyuan Zhou 19/10/9 9:29 PM
已删除: Currently the detailed physi... [17]

Siyuan Zhou 19/10/9 9:29 PM
已上移 [7]: 2010), which needs num... [18]

Siyuan Zhou 19/10/9 9:29 PM
已删除: ... [19]

Siyuan Zhou 19/10/9 9:29 PM
已上移 [8]: Conclusions

Siyuan Zhou 19/10/9 9:29 PM
已上移 [10]: The different relations... [22]

Siyuan Zhou 19/10/9 9:29 PM
已删除: Based on two indicators tha... [20]

Siyuan Zhou 19/10/9 9:29 PM
已上移 [9]: The quantitative differ... [21]

Siyuan Zhou 19/10/9 9:29 PM
已删除: The absorbing aerosols (BC... [23]

Siyuan Zhou 19/10/9 9:29 PM
已上移 [11]: By comparing the cha... [24]

Siyuan Zhou 19/10/9 9:29 PM
已上移 [12]:

Siyuan Zhou 19/10/9 9:29 PM
已删除: Comparing the influence of... [25]

Siyuan Zhou 19/10/9 9:29 PM
已删除: concurrent

Siyuan Zhou 19/10/9 9:29 PM
已删除: and attempted to address the causes

Siyuan Zhou 19/10/9 9:29 PM
已删除: limitation on studying aero... [26]

[revised manuscript text omitted]

Siyuan Zhou 19/10/9 9:29 PM
已删除: Altitudes

Siyuan Zhou 19/10/9 9:29 PM
已删除: ) and selected stations (dots

Siyuan Zhou 19/10/9 9:29 PM
已设置格式: 居中

Siyuan Zhou 19/10/9 9:29 PM
已删除: . ... [28]

[Figure]

.413

Figure 3. PDF of start time (units: LST), peak time (units: LST), duration (units: hours) and intensity (units: 0.1mm/hour) of heavy rainfall (data from CMA) on selected clean (blue lines) and polluted (red lines) conditions, respectively using indicator of (a) AOD and (b) CDNC (cm⁻³), during early summers from 2002 to 2012.

[Figure]

Figure 4. PDF of (a) start time (units: LST), (b) peak time (units: LST), and (c) duration (units: hours) of heavy rainfall on the days with SAI more than 75th percentile (blue lines, data from OMI) and days with AAI more than 75th percentile (red lines, data from OMI), during early summers from 2005 to 2012.

Siyuan Zhou 19/10/9 9:29 PM
已删除: 3

[Figure]

**(a) Percentage of BC AOD**    **(b) Percentage of sulfate AOD**

Figure 5. Percentages of AOD for (a) BC and (b) sulfate from MACC reanalysis data in summers (June – August) during 2002 to 2012.

[Figure]

Figure 6. PDF of start time (units: LST), peak time (units: LST) and duration (units: hours) of heavy rainfall on the different conditions of (a) BC and (b) sulfate. Blue/red lines stand for the condition of less/more BC or sulfate (AOD of BC or sulfate less than 25[th] /more than 75[th] percentile, data from MACC) during early summers from 2003 to 2012.

.443

.444

Siyuan Zhou 19/10/9 9:29 PM
已删除： . ... [30]

**(a) PDF with more/less SH**

[Figure]

**(b) Scatter distribution using AOD**

**(c) Scatter distribution using CDNC**

.445

.446 Figure 7. (a) PDF of start time (units: LST), peak time (units: LST), and duration (units: hours) of heavy

.447 rainfall with less moisture (blue lines, SH at 850 hPa less than 25th percentile, data form ERA-interim) and

.448 more moisture (red lines, SH at 850 hPa more than 75th percentile, data form ERA-interim). (b) and (c) are

.449 scatter distributions of SH-start time/peak time/duration for clean cases (blue points) and polluted cases (red

.450 points) respectively using AOD and CDNC. Green lines stands for the start/peak time at 8:00 LST or the

.451 duration is 0 hours. Positive (negative) values stand for the hours away from 8:00 LST or 0 hours in clean

.452 (polluted) cases. Blue (red) lines stand for the mean values of rainfall characteristics at each integer of SH in

.453 clean (polluted) cases.

.454

Siyuan Zhou 19/10/9 9:29 PM
已删除：6

.458

Siyuan Zhou 19/10/9 9:29 PM
已设置格式: 居中
Siyuan Zhou 19/10/9 9:29 PM
已删除： ... [31]

[Figure]

.459

.460

.461 Figure 8. PDF of start time (units: LST), peak time (units: LST), and duration (units: hours) of heavy rainfall

.462 on selected clean (blue lines) and polluted (red lines) conditions with SH at 850 hPa (from ERA-interim) less

.463 than 75th percentile, respectively using indicator of (a) AOD and (b) CDNC (cm$^{-3}$), during early summers from

.464 2002 to 2012.

.465

.466

.467

.468

.469

[Figure]

.472

.473 Figure 9. PDF of CF (units: %), CTP (units: hPa), COT (liquid and ice, units: none), CWP (liquid and ice,

.474 units: g/m²) and CER (liquid and ice, units: μm) on selected clean (blue dash lines: AOD<25th percentile; blue

.475 solid lines: CDNC<25th percentile) and polluted (red dash lines: AOD>75th percentile; red solid lines:

.476 CDNC>75th percentile) heavy rainfall days. All cloud variables are obtained from MODIS C6 cloud product.

.477

.478

.479

.480

[Figure]

.481

.482 Figure 10. PDF of CF (units: %, data from MODIS) respectively for the conditions of less BC/sulfate (blue

.483 lines, AOD of BC/sulfate less than 25th percentile, data from MACC) and more BC/sulfate (red lines, AOD of

.484 BC/sulfate more than 75th percentile, data from MACC) cases with heavy rainfall during 10 early summers

.485 (2003-2012).

Siyuan Zhou 19/10/9 9:29 PM
已删除: 8
Siyuan Zhou 19/10/9 9:29 PM
已删除: %)

Figure 11. A schematic diagram for aerosol impacts on heavy rainfall over Beijing-Tianjin-Hebei region.

Siyuan Zhou 19/10/9 9:29 PM
已删除: 9

---

## Author Response (AR5)

**Responses to Comments from Reviewer and Editor**

**Thanks for the comments and suggestions. The point-by-point response is given below, in bold.**

*The discussion section could be moved earlier in the paper (before the conclusions section) - this would be a more usual structure and would help give the conclusions more prominence.*
**R: Agree, it is moved.**

*Following my previous comment on calculating the CDNC, the gridbox mean values are not really suitable for calculating the CDNC as the calculation is highly non-linear. Doing the calculation using the joint histogram does not require that much more work, but will produce better results. This may not change the results much, but it might be a good idea for future studies at the very least. I have attached a plot of the mean CDNC in March-April-May over the study region, calculated from the MODIS L3 histograms for the period 2006-2010 (the data I had easily available). The plot on the left shows the mean CDNC across the nearby region.*
**R: Thanks. We very much appreciate the illustrated plot. We have followed the suggestion using the joint histogram data; see Section 2.1.3 and all the CDNC values and relevant tables/figures have been changed. As pointed out by the Reviewer, the main conclusions remain.**

*The English in the paper could still use some improvement, although the meaning is usually clear. I have not gone through the paper again to highlight all the changes, but they seem to be primarily in the new sections.*
**R: We have polished the manuscript.**

*Other comments:*
*L307 - The SH is defined correctly earlier as the specific humidity, rather than the ``water vapor content"*
**R: The sentence is deleted.**

*L434 - The cases of heavy rainfall using CDNC seem more extreme - then at L440 - the change in rainfall intensity is not significant. Please clarify this*
**R: The "extreme" did not mean the rainfall intensity, but the bigger difference between clean and polluted cases using CDNC. Since we re-calculated the CDNC, and accordingly this sentence is now meaningless and has been deleted.**

*L639 - 'accumulate more water in the cloud' - Is this a reference to some kind of Albrecht (1989) type effect? If so, it might be good to state it (although an earlier start to precipitation does not match this). If not, it would also be good to state it (and explain what it is).*

**R: Our results indicate that the CF, COT and CWP become larger when AOD/CDNC increases (Sect. 4.1) and that the liquid CER becomes larger when moisture increases in the polluted environment (Sect. 4.2), we suggest that more aerosols can serve as CCN, which in a moisture sufficient environment can hold more liquid water in the cloud. Therefore, it is not the same as the second indirect effect on cloud extent and lifetime as in Albrecht (1989), which states that, for a fixed liquid water path, the increased CCN would lead to smaller cloud particle sizes and thus suppression of precipitation and prolonging of the cloud lifetime. The sentence is modified accordingly, see L410-412 and L468-470.**

*L654(and associated section) - CDNC is not CCN. As the editor has pointed out, in an updraft limited regime, variations in CDNC come primarily from changes in updraft. Just stating that you are using CDNC to represent CCN does not make CDNC actually represent CCN.*

**R: Yes, it is re-written, see L427 and L432.**

*L694 - Following from my comments in previous reviews - The relationship is not completely meaningless, just difficult to interpret. Also, this sentence does not make much sense to me - you state that this comparison cannot be done, then use it to infer the role of aerosols. I would suggest you leave this sentence out, you have plenty of other results without relying on this.*

**R: Agree, the sentence is deleted.**

*L853 - It is not clear the large scale updraft is a good measure of the in-cloud updraft relevant for droplet formation.*

**R: Agree, we have modified the statement, see L550-553.**

[revised manuscript text omitted]

Siyuan 20/2/19 11:34 PM
已删除: associated

Siyuan 20/2/19 11:34 PM
已删除: supply

Siyuan 20/2/19 11:34 PM
已删除: effects

Siyuan 20/2/19 11:34 PM
已删除: on

Siyuan 20/2/19 11:34 PM
已删除: and

Siyuan 20/2/19 11:34 PM
已删除: and compared them with the effects of aerosols

Siyuan 20/2/19 11:34 PM
已删除: using AOD and CDNC

diurnal variation along with different types of aerosols, and the influence of moisture on the relationship between aerosols and heavy rainfall. Section 4 describes the concurrent changes of cloud properties associated with aerosols and compares the possible influences of CCN (represented by CDNC) and moisture (represented by SH) on the cloud properties. Section 5 gives the hypothesis about the mechanisms of aerosol effects on the heavy rainfall. Discussion and conclusions will be given in Sect. 6.

**2. Approach**

**2.1 Data**

Four types of datasets from the year 2002 to 2012 (11 years) are used in this study, which include (1) precipitation, (2) aerosols, (3) clouds, and (4) other meteorological fields.

**2.1.1 Precipitation**

To study the diurnal variation of heavy rainfall, the gauge-based hourly precipitation datasets are used, which were obtained from the National Meteorological Information Center (NMIC) of the China Meteorological Administration (CMA) (Yu et al., 2007) at 2420 stations in China from 1951 to 2012. The quality control made by CMA/NMIC includes the check for extreme values (the value exceeding the monthly maximum in daily precipitation was rejected), the internal consistency check (wiping off the erroneous records caused by incorrect units, reading, or coding) and spatial consistency check (comparing the time series of hourly precipitation with nearby stations) [Shen et al., 2010]. Here we chose 176 stations in the plain area of BTH region that are below the topography of 100 meter above sea level as shown in Fig.1, because we purposely removed the probable orographic influence on the rainfall diurnal variation, which is consistent with our previous work (Zhou et al., 2018). The record analyzed here is the period of 2002 to 2012. We selected heavy rainfall days when the hourly precipitation amount is more than 8.0 mm/hour (defined by *Atmospheric Sciences Thesaurus, 1994*). Here "a day" is counted from 8 LST to 8 LST next day (0 UTC to 24 UTC).

**2.1.2 Aerosols**

In this study, we used two satellite data and one reanalysis data to investigate the aerosol optical amount and distinguish the different aerosol types.

AOD is a proxy for the optical amount of aerosol particles in a column of the atmosphere and serves as the macro indicator for the division of aerosol pollution condition in this study, which was obtained from MODIS (Moderate Resolution Imaging Spectroradiometer) Collection 6 Level-3 aerosol product with the horizontal resolution of 1°x1° onboard the Terra satellite (Tao et al., 2015). The quality assurance of marginal or higher confidence is used in this study. The reported uncertainty in MODIS AOD data is on the order of (-0.02-10%), (+0.04+10%) (Levy et al., 2013). The Terra satellite overpass time at the equator is around 10:30 local solar time (LST) in the daytime, and the satellite data is almost missing when it is rainy during the overpass time.

Siyuan 20/2/19 11:34 PM 已删除: comparison

Siyuan 20/2/19 11:34 PM 已删除: heavy rainfall behaviors influenced respectively by

Siyuan 20/2/19 11:34 PM 已删除: and

Siyuan 20/2/19 11:34 PM 已删除: (CCN

Siyuan 20/2/19 11:34 PM 已删除: Conclusions

Siyuan 20/2/19 11:34 PM 已删除: discussion

Siyuan 20/2/19 11:34 PM 已删除: L3

[revised manuscript text omitted]

Siyuan 20/2/19 11:34 PM
已删除: might

Siyuan 20/2/19 11:34 PM
已删除: and accumulate

Siyuan 20/2/19 11:34 PM
已删除: thus increase the CF, COT and CWP.

Siyuan 20/2/19 11:34 PM
已删除: which denotes the decrease of the cloud top height, might

Siyuan 20/2/19 11:34 PM
已删除: probably

Siyuan 20/2/19 11:34 PM
已删除: also might result from

Siyuan 20/2/19 11:34 PM
已删除: heavy rainfall

Siyuan 20/2/19 11:34 PM
已删除: changes

Siyuan 20/2/19 11:34 PM
已删除: changes

Siyuan 20/2/19 11:34 PM
已删除: detailed

Siyuan 20/2/19 11:34 PM
已删除: grow

Siyuan 20/2/19 11:34 PM
已删除: 68.58

Siyuan 20/2/19 11:34 PM
已删除: 68.56

Siyuan 20/2/19 11:34 PM
已删除: 3

Siyuan 20/2/19 11:34 PM
已删除: 8

Siyuan 20/2/19 11:34 PM
已删除: can

Siyuan 20/2/19 11:34 PM
已删除: remains

Siyuan 20/2/19 11:34 PM
已删除: made

Siyuan 20/2/19 11:34 PM
已删除: significant test

Siyuan 20/2/19 11:34 PM

[revised manuscript text omitted]

Siyuan 20/2/19 11:34 PM

| Clean/Polluted | | CF |
|---|---|---|
| AOD | Clean | 62.8 (17.6) |
| | Polluted | 89.3 (12.9) |
| CDNC | Clean | 94.5 (6.1) |
| | Polluted | 97.4 (4.2) |

已删除:

Siyuan 20/2/19 11:34 PM

| Group (case number) | | CF |
|---|---|---|
| 1 | Clean, dry (153) | 93.8 (6.1) |
| 2 | Polluted, dry (128) | 95.6 (5.1) |
| 3 | Clean, wet (155) | 92.7 (7.0) $p_{1,3}>0.05$ |
| 4 | Polluted, wet (194) | 97.8 (4.4) |

已删除:

Siyuan 20/2/19 11:34 PM

已删除: has passed the significance test of 90% but

Siyuan 20/2/19 11:34 PM

已删除: %, and "P>0.1" stands for the difference did not pass the significance test of 90

.223
.224
.225
.226
.227
.228
.229
.230
.231
.232
.233
.234
.235
.236
.237
.238
.239
.240
.241

**Figures**

.242

.243

[Figure]

.244

Figure 1. Selected rainfall stations (blue dots) and topography (shading, units: m) in the BTH region (red box,
36–41° N, 114–119° E).

.245
.246
.247

.248

[Figure]

[Figure]

.249

.250 Figure 2. PDF of (a) AOD and (b) CDNC (cm$^{-3}$) (data from MODIS) on non-rainfall days (black lines),

.251 rainfall days (blue lines) and heavy rainfall days (red lines) in southwesterly during early summers from 2002

.252 to 2012. Numbers in the legends denote the sample number.

.253

.254

.255

[Figure]

[Figure]

.256

.257 Figure 3. PDF of start time (units: LST), peak time (units: LST), duration (units: hours) and intensity (units:

.260 0.1mm/hour) of heavy rainfall (data from CMA) on selected clean (blue lines) and polluted (red lines)

.261 conditions, respectively using indicator of (a) AOD and (b) CDNC (cm$^{-3}$), during early summers from 2002 to

.262 2012.

.263

[Figure]

.264

.265 Figure 4. PDF of (a) start time (units: LST), (b) peak time (units: LST), and (c) duration (units: hours) of

.266 heavy rainfall on the days with SAI more than 75th percentile (blue lines, data from OMI) and days with AAI

.267 more than 75th percentile (red lines, data from OMI), during early summers from 2005 to 2012.

.268

.269

[Figure]

.270

.271 Figure 5. Percentages of AOD for (a) BC and (b) sulfate from MACC reanalysis data in summers (June –

.272 August) during 2002 to 2012.

.273

.274

.275

[Figure]

.276
.277 Figure 6. PDF of start time (units: LST), peak time (units: LST) and duration (units: hours) of heavy rainfall
.278 on the different conditions of (a) BC and (b) sulfate. Blue/red lines stand for the condition of less/more BC or
.279 sulfate (AOD of BC or sulfate less than 25[th] /more than 75[th] percentile, data from MACC) during early
.280 summers from 2003 to 2012.
.281
.282

[Figure]

[Figure]

Siyuan 20/2/19 11:34 PM

已删除:

.283

.284 Figure 7. (a) PDF of start time (units: LST), peak time (units: LST), and duration (units: hours) of heavy

.285 rainfall with less moisture (blue lines, SH at 850 hPa less than 25[th] percentile, data form ERA-interim) and

.286 more moisture (red lines, SH at 850 hPa more than 75[th] percentile, data form ERA-interim). (b) and (c) are

.287 scatter distributions of SH-start time/peak time/duration for clean cases (blue points) and polluted cases (red

.288 points) respectively using AOD and CDNC. Green lines stands for the start/peak time at 8:00 LST or the

.289 duration is 0 hours. Positive (negative) values stand for the hours away from 8:00 LST or 0 hours in clean

.290 (polluted) cases. Blue (red) lines stand for the mean values of rainfall characteristics at each integer of SH in

.291 clean (polluted) cases.

.292

.293

[Figure]

[Figure]

Siyuan 20/2/19 11:34 PM

已删除：

Figure 8. PDF of start time (units: LST), peak time (units: LST), and duration (units: hours) of heavy rainfall on selected clean (blue lines) and polluted (red lines) conditions with SH at 850 hPa (from ERA-interim) less than 75[th] percentile, respectively using indicator of (a) AOD and (b) CDNC (cm$^{-3}$), during early summers from 2002 to 2012.

[Figure]

[Figure]

Siyuan 20/2/19 11:34 PM

已删除:

Figure 9. PDF of CF (units: %), CTP (units: hPa), COT (liquid and ice, units: none), CWP (liquid and ice, units: g/m$^2$) and CER (liquid and ice, units: μm) on selected clean (blue dash lines: AOD<25th percentile; blue solid lines: CDNC<25th percentile) and polluted (red dash lines: AOD>75th percentile; red solid lines: CDNC>75th percentile) heavy rainfall days. All cloud variables are obtained from MODIS C6 cloud product.

[Figure]

Figure 10. PDF of CF (units: %, data from MODIS) respectively for the conditions of less BC/sulfate (blue lines, AOD of BC/sulfate less than 25th percentile, data from MACC) and more BC/sulfate (red lines, AOD of BC/sulfate more than 75th percentile, data from MACC) cases with heavy rainfall during 10 early summers (2003-2012).

.322

.323

.324

.325

Figure 11. A schematic diagram for aerosol impacts on heavy rainfall over Beijing-Tianjin-Hebei region.

.326

.327

.328

.329